# Rotary Masked Autoencoders are Versatile Learners

**Uros Zivanovic**[1], **Serafina Di Gioia**[2, 3], **Andre Scaffidi**[3], **Martín de los Rios**[3], **Gabriella Contardo**[7, 3], and **Roberto Trotta**[3, 4, 5, 6]

[1]University of Trieste, Italy
[2]Abdus Salam International Centre for Theoretical Physics (ICTP), Italy
[3]Scuola Internazionale Superiore di Studi Avanzati (SISSA), Italy
[4]INFN – National Institute for Nuclear Physics, Italy
[5]ICSC - Centro Nazionale di Ricerca in High Performance Computing, Italy
[6]Imperial College London, United Kingdom
[7]University of Nova Gorica, Slovenia

## Abstract

Applying Transformers to irregular time-series typically requires specializations to their baseline architecture, which can result in additional computational overhead and increased method complexity. We present the Rotary Masked Autoencoder (RoMAE), which utilizes the popular Rotary Positional Embedding (RoPE) method for continuous positions. RoMAE is an extension to the Masked Autoencoder (MAE) that enables interpolation and representation learning with multidimensional continuous positional information while avoiding any time-series-specific architectural specializations. We showcase RoMAE's performance on a variety of modalities including irregular and multivariate time-series, images, and audio, demonstrating that RoMAE surpasses specialized time-series architectures on difficult datasets such as the DESC ELAsTiCC Challenge while maintaining MAE's usual performance across other modalities. In addition, we investigate RoMAE's ability to reconstruct the embedded continuous positions, demonstrating that including learned embeddings in the input sequence breaks RoPE's relative position property.

## 1 Introduction

The framework offered by foundation models has shifted the machine learning landscape by establishing new benchmarks on a variety of modalities and tasks. Specifically, Transformers [58] have achieved state-of-the-art performance across many domains, from vision [15] to natural language processing [61]. Given the ability of Transformers to handle sequential data such as natural language, they naturally became appealing for time-series, which arise in a large variety of domains, including health, finance and astrophysics. Such data can often be irregularly sampled in the temporal dimension. Being originally designed for sequences of text, the base Transformer architecture is not able to deal with such irregularly sampled data, by default only supporting quantized positional information as is found in natural language. This lack of support for continuous positional information becomes a limitation when extending Transformers to other modalities, degrading performance on tasks requiring precise temporal modelling and hindering the model's ability to capture complex patterns in non-uniformly sampled time-series.

Various specializations to the Transformer have been proposed to overcome this limitation. These can be divided into two main types: modifications of the internal architecture of the Transformer (e.g. modifying the feedforward layer in the Transformer Encoder Block [19, 46]) and novel positional embeddings (e.g. using a neural ODE [12]). Alternatively, modern state space models

like Mamba [22] or S5 [51] are natively able to model various modalities such as text and images in addition to irregular time-series. Extending the capability of Transformers to irregular time-series while staying within existing frameworks developed for fields such as Natural Language Processing (NLP) and computer vision would allow one to easily benefit from ongoing developments within the Transformer "ecosystem".

To this end, we propose a new representation learning method utilizing Rotary Positional Embeddings (RoPE) [54] for continuous position in combination with Masked Autoencoder (MAE) [23] pre-training: the Rotary Masked Autoencoder (RoMAE). We investigate the performance of this framework on a variety of tasks with different modalities. RoMAE obtains highly competitive results when compared to specialized, state-of-the-art approaches for individual tasks while maintaining MAE's excellent performance in computer vision and audio. RoMAE is therefore extremely versatile while being built up from only standard Transformer methods.
Our contributions are threefold:

1. **Continuous Positional Embedding with RoPE:** We investigate how RoPE can be used to embed continuous positions, expanding on its original concept introduced in the *RoFormer* architecture [54], which did not address non-uniform or real-valued timestamps. We show that learned input embeddings can invalidate RoPE's relative distance guarantees, and we provide new empirical insights into RoPE's ability to embed varying positional scales.

2. **RoMAE**: An expansion of MAE that works natively with irregular multivariate time-series without sacrificing any performance on standard modalities such as images and audio. Utilizing standard off-the-shelf methods developed for Transformers in NLP and Computer Vision, RoMAE shows that a specialized architecture is not required to achieve strong performance on irregular time-series.

3. **Experimental Analysis**: We compare RoMAE with state-of-the-art deep learning (DL) models, conducting experiments on the following tasks and modalities: (i) irregularly sampled multivariate time-series classification, (ii) image classification, (iii) irregularly sampled time-series interpolation and (iv) audio classification.

This work is structured as follows: Section 2 covers related works on MAE, RoPE, and irregular time-series. Section 3 provides the necessary background material. Section 4 details the workings of RoMAE and establishes the theoretical framework we use to tackle a variety of modalities. Section 5 presents the results of our experiments. Section 6 discusses our results. Section 7 concludes the paper.

## 2   Related Work

**Rotary Positional Embeddings**: RoPE was initially proposed in RoFormer [54] as a simple and effective Relative Position Embedding (RPE) method that is independent from the Multi-Head Attention (MHA) implementation being used. Despite RoPE encoding relative position, models incorporating RoPE have been shown not to generalize to sequences longer than the ones shown during training. To improve sequence length extrapolation, various works have proposed increasing RoPE's base wavelength from 10 000 up to 500 000 [63, 43, 62]. Alternatively, YaRN [38] proposes to interpolate RoPE's frequencies $\theta_i$ during inference to avoid out-of-distribution angles. There has also been recent discussion on the usefulness of the long-term decay property of RoPE [4]. In this work, we provide additional experimental evidence supporting the argument against the long-term decay of RoPE.

**RoPE in Vision Transformers**: 2D Hand-crafted and learned Absolute Positional Embedding (APE) methods have both been shown to give similar performance on Computer Vision benchmarks when used with Vision Transformers (ViT) [15]. Later works [25, 34] have shown the impact of RoPE on multi-resolution inference, finding that RoPE improves ViT's extrapolation performance. To extend RoPE to 2D, Axial RoPE [17] applies RoPE twice, once for each dimension. Additional learnable parameters can be added to Axial RoPE to encode diagonal positional information as well [25]. Taking advantage of RoPE's independence from the specific Multi-Head Attention (MHA) implementation, Vision X-former [29] (ViX) is a variant of ViT that utilizes RoPE with linear MHA implementations, making it more computationally efficient. Overall, adding RoPE to ViT has been shown to be a beneficial change, improving model performance, and enabling extrapolation to higher resolutions during inference.

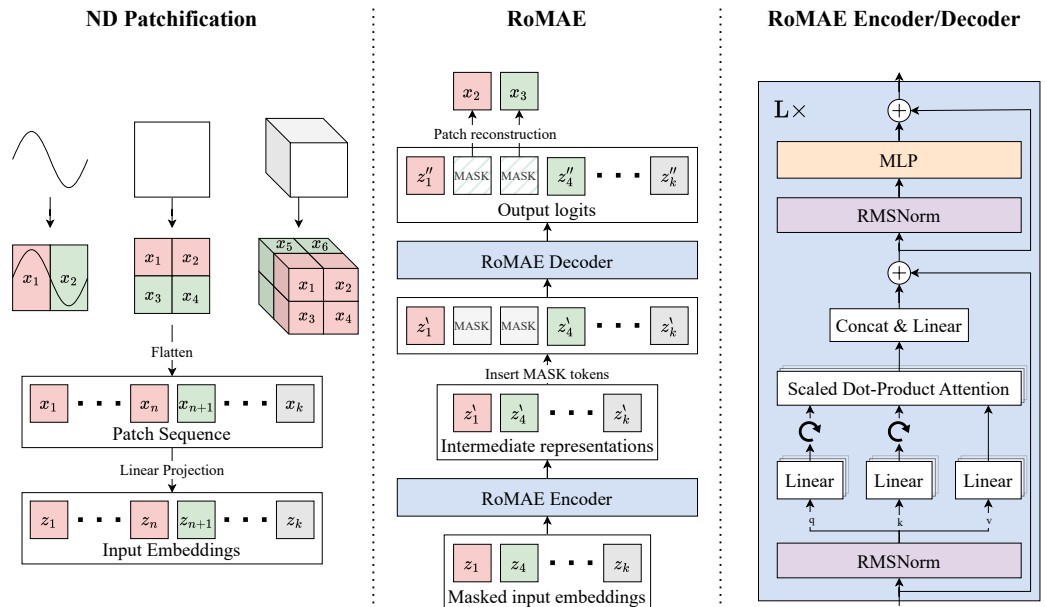

Figure 1: Overview of the RoMAE pipeline. **Left**: Visualization of data embedding via multi-dimensional (ND) patchification for illustrative data realisations in 1, 2 and 3D. **Centre**: Full depiction of RoMAE architecture. The optional `[CLS]` token is omitted from the input sequence for simplicity. **Right**: The RoMAE encoder/decoder with RoPE operations denoted by rotational arrows.

**Masked Autoencoders**: Architectures such as BERT [14] and GPT [40, 41, 6] have shown that self-supervised pre-training tasks greatly boost the downstream fine-tuning performance of Transformers in NLP. MAE [23] is an approach that adapts BERT's masked modelling pre-training task to images. Although MAE is not the first work to conduct masked image modelling, it is one of the most widely used methods. MAE has been shown to be both data-efficient [57] and scalable [23, 59]. It has also been adapted successfully to a variety of modalities other than images such as video [57, 18], audio [28], and point clouds [55]. MAE has also been combined with RoPE in MAETok [8], which uses the trained model as a tokenizer for diffusion models. Although MAE has been used in a variety of contexts, we highlight that many of these contexts have required task-specific specializations to the backbone Transformer architecture, unlike the approach we present here.

**Transformers for Irregular Time-series**: Adapting the Transformer architecture for irregular time-series is a long-standing research topic with many methods having been proposed [60]. When tokenizing the input, a popular approach is to insert the time for each point through positional embeddings – an approach recently used by models such as ContiFormer [12], Timer [33], and the concurrent work TrajGPT [52], which, similar to RoMAE, uses RoPE for the task. Alternatively, one can also convert the time-series data into 2D images and process them using an off-the-shelf ViT [32]. As a pre-training task, methods focus either on autoregressive trajectory prediction [52, 33], or BERT-style interpolation through masked modelling [37]. MAE has not been used for irregular time-series pre-training yet.

## 3 Background

**Attention:** We follow the formulation for Attention proposed in the original Transformer [58]. Let $\mathbf{z}$ be a sequence of $k$ embeddings $z_i \in \mathbb{R}^{d_{\text{model}}}$ for $i \in [1, 2, ..., k]$ and $d_{\text{model}}$ be the dimensionality of the model. Three linear layers are applied to transform $\mathbf{z}$ into sequences containing queries ($q_i$), keys ($k_i$), and values ($v_i$). E.g., $q_i = W_q z_i$, $k_i = W_k z_i$, $v_i = W_v z_i$. The matrices $Q, K, V$ containing $q,k,v$ are then passed through Scaled Dot-Product Attention (SDPA) as defined in Equation (1):

$$\text{Attention}(Q, K, V) = \text{softmax}\left(\frac{QK^T}{\sqrt{d_{\text{model}}}}\right)V \tag{1}$$

The key operation in SDPA is the dot product between $q_i$ and $k_i$, which determines how the values $V$ will be mixed with one another. Because Attention is permutation-invariant, positional information must be encoded within $q_i$ and $k_i$ to allow SDPA to reason about position.

**Regular and Irregular Dimensions:** RoMAE is designed to work with both regular and irregular dimensional data. Specifically, we consider inputs of the form: $\mathbf{x} \in \mathbb{R}^{d_1 \times d_2 \times \cdots \times d_D}$ where $D$ is the number of dimensions in $\mathbf{x}$ and $d_i$ is the size of dimension $i$ for $i \in [1, \cdots, D]$.

**Definition 3.1** (Regular and irregular dimensions). A *regular* dimension is one where all points are equally spaced, while an *irregular* dimension is one where the distance between points varies.

Some dimensions in the input $\mathbf{x}$ may be irregular while others could be regular. For example, a set of images sampled at irregular times has height and width as two regular dimensions and time as one irregular dimension.

### 3.1 Rotary Positional Embeddings

Given input $x_m \in \mathbb{R}^{d_x}$ with position $m$ and even dimensionality $d_x$, RoPE partitions $x_m$ into disjoint 2D subspaces $x_m^{(i)}$ with $i \in [1, 2, ..., d_x/2]$, rotating each subspace as:

$$\begin{pmatrix} \cos \ m\theta_i & -\sin \ m\theta_i \\ \sin \ m\theta_i & \cos \ m\theta_i \end{pmatrix} x_m^{(i)} \tag{2}$$

The $\theta_i$ values are generated in the same way as for sinusoidal positional embeddings [58]: $\theta_i = 10000^{-2(i-1)/d_x}$. Therefore, each subspace is rotated by a different amount depending on the individual $\theta_i$. RoPE is applied directly to the queries and keys before they enter SDPA. Because the dot product relies only on the angle between two vectors and their magnitude, RoPE is a RPE method.

$p$-**RoPE:** In this work we make use of $p$-RoPE [4], a truncated version of RoPE where only the $p$ percent of smallest $\theta_i$ values are kept. This cuts out a fraction $(1 - p)$ of the frequencies in RoPE and thus leaves a portion of the input embedding space unchanged. The unchanged region in the embedding space provides the model with a data channel it can use to pass information into SDPA without any modifications by RoPE, making the model more robust to varying sequence length. We use a value $p = 0.75$, which has been shown by Barbero et al. [4] to work well.

**Axial RoPE:** To encode multi-dimensional position, we utilize Axial RoPE [17]. In Axial RoPE, the input is split into $D$ subspaces of size $d_{\text{model}}/D$, where $D$ is the number of positional dimensions. Then we apply $p$-RoPE to each subspace, encoding the positional value for that dimension. We note that since RoPE requires that embeddings be even and Axial RoPE requires that embeddings be divisible by $D$, this puts constraints on the possible values that $d_{\text{model}}$ can take.

## 4  Method

An overview of RoMAE is shown in Figure 1.

**N-Dimensional (ND) Patchification:** Given inputs $\mathbf{x}$ as described in Section 3, we define a patch size $(p_1, \cdots, p_D)$ and divide each dimension into $N_i = d_i/p_i$ non-overlapping segments, where $d_i$ is the size of dimension $i$. These are flattened, creating a sequence of patches with length $k = \prod_{i=1}^{D} N_i$ and number of elements per patch $n_p = \prod_{i=1}^{D} p_i$. Finally, a linear layer $W^{n_p \times d_{\text{model}}}$ is applied to each patch to project it into the embedding dimension $d_{\text{model}}$. This process is illustrated in Figure 1 and is the same as employed by ViT [15] for images and ViViT [1] for video. It can also be understood as using non-overlapping $N$-dimensional convolutions with the number of channels equal to $d_{\text{model}}$. After ND-patchification, we have a sequence of embeddings $\mathbf{z}$, which can be passed into the RoMAE Encoder.

**Proposition 4.1.** For any *irregular* dimension $d_i$ in $\mathbf{x}$, the corresponding patch size for that dimension $p_i$ must be equal to 1.

**Discussion:** This limitation emerges from the requirement that each patch has the same number of data points $n_p$ inside it. Note that mixing irregular and regular dimensions is not an issue. E.g., for an

irregularly-sampled time-series of images, one could choose a patch size of (1, 16, 16) for the time, height, and width respectively.

Because the ND-patchification process flattens all dimensions into a single sequence, RoMAE is able to jointly model and attend to all patches across all dimensions at once. The drawback of this is that the number of tokens grows exponentially with the number of dimensions. In this work we only utilize this process up to $D = 3$. For highly irregular multivariate time-series, we also utilize a different approach that is described in Section 4.2.

## 4.1 Overall Structure

RoMAE's structure follows MAE's, using an asymmetric encoder/decoder, with the encoder being much larger than the decoder. Although both the encoder and decoder in RoMAE are a Transformer Encoder similar to BERT [14], we bring over recent developments in NLP from models such as Llama [43]. Specifically, we utilize the popular Sigmoid Linear Unit [24, 16] for the non-linearity. We also use RMSNorm [66] instead of Layer Normalization [2], as it has been shown to be more computationally efficient while maintaining the same level of model convergence. For architectural details, including definitions of various model sizes, we refer to Appendix A.1 and Appendix A.2.

**Pre-Training Task:** Given a sequence of input patches, we uniformly mask a percentage of them. After projecting the unmasked patches into embeddings, a learned [CLS] token is optionally appended to the start of the sequence. This token becomes useful during fine-tuning, when an MLP head can be placed on top of it to conduct classification. The embeddings are then passed through the RoMAE Encoder to create an intermediate representation $\mathbf{z}'$. A set of learnable [MASK] tokens is then appended to $\mathbf{z}'$, with each [MASK] token receiving the positional information corresponding to a patch that was masked out. This sequence is then passed through the RoMAE Decoder, after which the model head predicts $n_p$ values for each patch that was masked out. After pre-training, the decoder is removed and the intermediate representations $\mathbf{z}'$ are used for downstream tasks.

## 4.2 Positional Information in RoMAE

**Continuous Axial RoPE:** Although RoPE is originally designed to be used with discrete positions such as those found in text, we observe that Equation (2) works with any $m \in \mathbb{R}$. We make use of this in RoMAE to encode continuous position. Specifically, alongside the input values $\mathbf{x}$, RoMAE also accepts a sequence $\mathbf{s} = [s_1, \cdots, s_k]$, $s_i \in \mathbb{R}^D$, containing the positional information for each patch. This is then applied within RoMAE using Axial RoPE as described in Section 3.1.

**Dealing With Many Irregular Dimensions:** Although we are able to encode multi-dimensional positional information using Axial RoPE, this does not scale well to a large number of dimensions due to having to divide $d_{\text{model}}$ by $D$. To overcome this, we optionally reserve a dimension in Axial RoPE that is used to store the dimensional index $i$ for $i \in [1, 2, \cdots, D]$. E.g., if an embedding belongs to dimension 4, it will receive a positional encoding of 4. When using this approach we include the learned [CLS] token, which allows the model to recover the dimensional index despite RoPE being a RPE method (as a consequence of Proposition 4.2). We utilize this method with the ELAsTiCC dataset in Section 5.4 which allows us to reduce the number of positional dimensions from 6 to 2.

## 4.3 Effects of Relative Position in RoMAE

Here we present an analysis of the effects of switching from absolute to relative position in RoMAE.

**Proposition 4.2** (Reconstructing absolute position). When a learned [CLS] token is included in the input sequence, the RoMAE Encoder is able to reconstruct the original set of positions $\mathbf{s}$ as described in Section 4.2.

**Intuition:** The [CLS] token provides the model with an "anchor". This allows the model to compare each input embedding to the [CLS] token, and reconstruct its absolute position. We provide a proof in Appendix C, as well as experimental evidence in Section 5.1.

**Corollary 4.1** (Translational invariance in the RoMAE Encoder). When no learned [CLS] token is included in the input sequence, positional information in the RoMAE Encoder is relative. This makes the RoMAE Encoder invariant to translations of the input embedding positions.

**Discussion:** Translational invariance in the RoMAE Encoder makes the pre-training task more difficult because the model cannot make predictions based on the overall position of a token in the input. E.g., with absolute position, the model can learn that objects of interest may often appear near the centre of an image.

**Corollary 4.2** (Effect of distance on absolute position reconstruction). RoMAE is able to recover the absolute position of any embedding $z_i$ with regard to Proposition 4.2, irrespective of the position $s_i$ of that embedding.

**Discussion:** A claimed property of RoPE is that it causes the dot product between queries and keys to decay as their positions grow further apart [54]. In Appendix B.1, we provide empirical evidence showing that RoMAE is able to reconstruct position almost perfectly over two different scales of distance. We also refer to the proof by Barbero et al. [4], showing that it is possible to construct a key for any non-zero query and any distance such that the softmax value in Attention is maximized at that distance.

Overall, relative position is a key element that influences the training dynamics of RoMAE. This allows us to investigate RoPE from a new angle, drawing new conclusions on the effect of learned embeddings and supporting prior claims that the long-term decay property is not significant to the functioning of RoPE.

## 5 Experiments

Throughout the experiments we make use of different sizes of RoMAE: RoMAE-tiny, RoMAE-small, and RoMAE-base, as detailed in Appendix A.1.

**Compute Details:**

The experiment on the Tiny ImageNet data set (Section 5.2) was run on one node of a Slurm cluster, utilizing two NVIDIA Tesla V100 GPUs for 5 hours.
The experiment on the DESC ELAsTICC Challenge[1] (Section 5.4), was run on a Slurm cluster using 4 nodes for $\sim 4$ hours with each having 4 Nvidia A100 (with 64GB memory) GPUs.
Together, the experiments on the UEA Time-Series Archive [3] (Section 5.4), Pendulum dataset [5] (Section 5.4), and absolute position experiments (Section 5.1) were run on a 1080ti GPU for a total of $\sim 1.5$ hours.
All interpolation experiments (Sec. 5.5) were run on a single NVIDIA A100-PCIE-40GB GPU (internal cluster), utilising $\leq 5$ GB memory, $\sim 10$min for the spirals dataset, $\sim 30$ mins for the synthetic dataset, and $\sim 3$ hours for PhysioNet.
Model and experimental code for RoMAE is made public through a convenient Python package.[2]

### 5.1 Reconstructing Absolute Position

To verify the model's ability to reconstruct absolute positional information according to Proposition 4.2, we give the model a sequence of 10 identical values as input. Each embedding is then given a 1D position sampled uniformly between 0 and 50. We then use the same linear head to predict the position for all tokens. Because the model dimension $d_{\mathrm{model}}$ has an effect on the number of $\theta_i$ values RoPE uses, we also test a variety of model sizes. We run each test 5 times and report the mean and standard deviation. Our generated training set has 20 000 samples while our generated test set has

---

[1]`https://portal.nersc.gov/cfs/lsst/DESC_TD_PUBLIC/ELASTICC/`
[2]`https://chromeilion.github.io/RoMAE-Website/`

Table 1: Position reconstruction MSE (mean $\pm$ std) for various sizes of RoMAE.

| Model size | With [CLS] | No [CLS] |
|---|---|---|
| RoMAE-tiny | 0.062 (0.007) | 200.33 (0.001) |
| RoMAE-small | 0.0057 (0.002) | 200.33 (0.001) |
| RoMAE-base | 0.0031 (0.002) | 200.33 (0.002) |

Table 2: Results on Tiny ImageNet across various versions of RoMAE-small

| Model | F-score ($\pm$ std) |
|---|---|
| RoMAE (no [CLS]) | 0.500 (0.011) |
| RoMAE ([CLS]) | 0.475 (0.006) |
| RoMAE (absolute) | 0.479 (0.010) |

4000. Reported Mean Squared Error (MSE) is the average MSE over the test set. Results are shown in Table 1.

We observe a clear difference between the model that uses the [CLS] token and the one that does not. When supplied with the learnable token, RoMAE is able to reconstruct the original absolute position almost perfectly. Larger models seem to achieve a better MSE, although all sizes perform well. For all experimental details and an additional experiment on absolute position reconstruction we refer to Appendix D.5 and Appendix B.1 respectively.

## 5.2 Tiny ImageNet

To investigate the effect of positional embedding and the learned [CLS] token on RoMAE, we train three versions of RoMAE on Tiny ImageNet [31]; RoPE with the [CLS] token, RoPE without the [CLS] token, and absolute sinusoidal positional embeddings [58] with the [CLS] token. When fine-tuning RoMAE without the [CLS] token, we place the classification head on top of the mean of the output embeddings, otherwise we place the head on top of the [CLS] token. The final configuration with absolute positional embeddings is very similar to MAE, making for a good comparison. We use a patch size of (16, 16) and mask 75% of the input, similar to MAE. After pre-training each model for 200 epochs, we fine-tune for another 15. We also follow the procedure outlined by MAE to compute the pre-training loss using normalized patch values instead of pixel values.

For full experimental details we refer to Appendix D.1. The results are shown in Table 2. Although all 3 models perform similarly, we highlight that RoMAE with RoPE and no [CLS] token performs slightly better than both RoPE with [CLS] and absolute positional embeddings. This could be because of the models translation invariance (Corollary 4.1). The difference could also come from placing the classification head on top of the mean of output embeddings instead of the [CLS] token. Overall, our results indicate that RoMAE performs at least as well as MAE on images.

## 5.3 Audio benchmark

Table 3: Results for the ESC-50 benchmark for audio datasets, for the RoMAE-small model.

| Model | Accuracy |
| --- | --- |
| SSAST (Librispeech) | 80.0 |
| RoMAE-small (Librispeech) | **83.2** |
| SSAST (AudioSet-20k) | 82.2 |
| RoMAE-small (AudioSet-20k) | **84.7** |

We chose to test RoMAE's ability to classify audio files, after a self-supervised pre-training on unlabeled audio datasets, inspired by the SSAST pretraining strategy [21]. SSAST is the first Vision transformer-based model that introduces a self-supervised pretraining strategy, supporting arbitrary patch size, for the audio representation learning.

As our pretraining datasets, we used a modified version of the Audioset [20] and Librispeech [36] datasets. AudioSet is a 2017 multi-label audio event classification dataset. Using a carefully structured hierarchical ontology of 635 audio classes guided by manual curation and expert knowledge, the authors collected data from human labelers to probe the presence of specific audio classes, including, for example, human sounds, animal sounds, music, natural sounds, in 2 million 10-second segments of YouTube videos. However, access to the original 2 million audio clips is fraught with difficulty, as a consistent subset of YouTube videos is no longer available. In order to conduct a reproducible experiment, we decided to use as training set the balanced training AudioSet-20k made available on Hugging Face.[3] We thus pretrain RoMAE using two different data sets: AudioSet-20k and the Librispeech dataset. For the Audioset dataset we used the training/validation split provided by the downloaded dataset, while for Librispeech, downloaded and preprocessed using the scripts provided on the SSAST Github repo, we used a 70/30 split.

In order to apply our model to the audio datasets, we first transform the audio waveforms to Mel spectrograms. First, the input audio waveform of length $t$ seconds is converted into a sequence

---

[3] `https://huggingface.co/datasets/agkphysics/AudioSet`

of 128-dimensional log-Mel filterbank[4] (fbank) features computed with a 25ms Hamming window every 10ms. This results in a $128 \times 100\,t$ spectrogram. For the pretraining step, we followed the patchification strategy adopted in [21] and we split the spectrogram into a sequence of $N$ ($16 \times 16$) patches, where $N = 12(100t - 16)/10$ is the number of patches and the effective input sequence length for the model. We refer to Appendix D.2 for the list of hyperparameters adopted for the pretraining and finetuning of the model. We would like to highlight here that the pretraining was run for 150 epochs, without the `[CLS]` token, while we choose to adopt a mask ratio equal to 0.75, that is very similar to the masking fraction associated with the SSAST-patch 400 model.

For the finetuning audio classification benchmark, we used the ESC-50 dataset [39], consisting of 2000 5-second environmental audio recordings classified into 50 classes. The current best results on ESC-50 are accuracies of 86.5 and 94.7 obtained with supervised training (on AudioSet-2M) respectively by SOTA-S and SOTA-P models. The SSAST model, which is the only ViT model adopting self-supervised training for this task, achieved accuracies of 82.2 and 84.6 when trained, respectively, on AudioSet-20K and AudioSet-2M (as reported in Table 2 of [21]), showing that the size and richness of the pretraining dataset impacts, in a non-negligible way, the performance of that model on the finetuning tasks. Since we did not have sufficient computational resources to pretrain RoMAE on AudioSet-2M, we chose to compare our model performance with that of the SSAST model pretrained on AudioSet-20K and Librispeech, respectively. We report the benchmark results in Table 3, showing that RoMAE performs better than the SSAST model in these two cases.

### 5.4 Irregular Time-series Classification

Table 4: Light curve classification results on ELAsTiCC. A ✓ or ✗ corresponds to whether the Alert Mask (AM) is used.

| Method | AM | F-score |
|---|---|---|
| Transformer [58, 7] | ✓ | 0.5256 |
| ATAT [7] | ✓ | 0.6270 |
| RoMAE-tiny-shallow | ✓ | 0.6649 |
| RoMAE-tiny-shallow | ✗ | 0.7106 |
| RoMAE-tiny | ✓ | 0.7205 |
| RoMAE-tiny | ✗ | **0.8029** |

Table 5: Regression MSE $\times 10^{-3}$ (mean $\pm$ std) on the Pendulum dataset. We use a custom RoMAE model size.

| Model | Regression MSE ($\times 10^{-3}$) |
|---|---|
| ODE-RNN [44] | 7.26 (0.41) |
| RKN-$\Delta_t$[5] | 5.09 (0.40) |
| ContiFormer [11] | 4.63 (1.07) |
| CRU [45, 51] | 3.94 (0.21) |
| S5 [51] | 3.41 (0.27) |
| RoMAE | **3.32 (0.13)** |

**DESC ELAsTiCC Challenge**    The DESC ELAsTiCC Challenge is a multivariate irregular time-series dataset consisting of $\sim$1.8M simulated light curves and 36 classes of astronomical objects. Each light curve consists of 6 irregularly sampled channels (called 'bands'). To embed this data in RoMAE, we follow the procedure described in Section 4.2. This results in a 2 dimensional positional embedding, where one dimension embeds the time, and the second embeds the channel index. Although ELAsTiCC has additional metadata for each light curve, we compare performance only across raw light curves. Some points in the light curve are marked through an alert mask as being unlikely to contain useful information. We evaluate RoMAEs performance both with these points and without. For more details on how these points are chosen we refer to Appendix D.6. We train all RoMAE models by conducting full pre-training for 200 epochs with a masking ratio of 50%, then fine-tuning for 25 epochs. for full details and a visualization of the data we refer to Appendix D.6.

Table 4 shows our results using two sizes of RoMAE, and compares with ATAT [7], a Transformer architecture specialized for ELAsTiCC. Despite the latter's specialization, RoMAE-tiny-shallow (with a comparable number of parameters as ATAT) improves over ATAT by about .04 F-score when using the alert mask. The larger RoMAE-tiny with the alert mask achieves an even greater improvement of about .1 F-score. A key reason for RoMAE's better performance might be that ATAT does not conduct any pre-training. In the case of RoMAE-tiny, the larger scale of the model also likely plays a role. We also find it notable that when ignoring the alert mask and using all points, RoMAE performs even better, suggesting that perhaps these points should not be ignored.

---

[4]The Mel Filterbank transform computes weighted averages of bins to provide spectral power estimates on a logarithmic frequency scale, which is more affine to human hearing resolution.

Table 6: Accuracy across various datasets from the UEA Time-series Archive.

| Dataset | Model | | | | |
|---|---|---|---|---|---|
| | TST [65] | mTAN [46] | S5 [51] | ContiFormer [11] | RoMAE-tiny |
| BM | 0.9667 | **0.9917** | 0.9833 | 0.9750 | **0.9917** |
| CT | 0.9742 | 0.9529 | 0.9610 | 0.9833 | **0.9882** |
| EP | **0.9589** | 0.9203 | 0.9074 | 0.9324 | 0.9517 |
| HB | 0.7398 | **0.7789** | 0.7333 | 0.7561 | 0.7447 |
| LSST | 0.5520 | 0.5307 | **0.6389** | 0.6004 | 0.6225 |

Table 7: Results for the interpolation experiments discussed in Section 5.5.

| Synthetic (MSE) | | Spirals (RMSE) | |
|---|---|---|---|
| HeTVAE [35] | **0.223 ± 0.070** | | |
| RoMAE-tiny | 0.233 ± 0.007 | Transformer [58] | 1.37 ± 0.14 |
| | | Latent ODE [10] | 2.09 ± 0.22 |
| PhysioNet (MSE) | | ContiFormer [11] | 0.49 ± 0.06 |
| HeTVAE [35] | 0.562 ± 0.022 | RoMAE-tiny | **0.0183 ± 0.007** |
| RoMAE-tiny | **0.467 ± 0.021** | | |

**UEA Multivariate Time-series Archive** We evaluate RoMAE on a variety of datasets from the UEA Multivariate Time-series Archive [3]. To make the datasets irregular we follow the procedure outlined by Kidger et al. [30], dropping 30% of the observations. Because all variates are present at each time-step, the only irregular dimension is time. Therefore, to embed this data in RoMAE, we combine all variates per time-step into one embedding. This is a much simpler setup than the one used for the ELAsTiCC dataset in Section 5.4. For each dataset we conduct pre-training for 400 epochs. When fine-tuning, we found it necessary to change hyper-parameters between different datasets. For more details on our experimental setup we refer to Appendix D.3.

Table 6 presents the mean accuracy from 3 full training runs (pretraining + finetuning), as well as published results from a variety of models. Overall, RoMAE-tiny performs similarly or better than the comparators, and is able to handle the various datasets without issues. All the datasets trained on are relatively small, with some being on the order of hundreds of samples. Therefore, this experiment also shows how data-efficient RoMAE can be.

**Irregular Time-series Regression: Pendulum Dataset** The Pendulum dataset [51] is an irregular time-series dataset consisting of irregularly sampled images of a pendulum. To embed the images in RoMAE, we use a patch size of $(1, 24, 24)$ for (time, height, width). This corresponds to 1 embedding per time-step/image. RoMAE is trained directly on regression without any pre-training, predicting the sine and cosine of the angle of the pendulum which follows a non-linear dynamical system. We use a custom size for RoMAE which contains only 2 layers and an MLP hidden size of 30. This is much smaller than RoMAE-tiny and provides for a fairer comparison with other models trained on this dataset. For additional information on the dataset and full experimental details we refer to Appendix D.4. After training on 20 different seeds, the mean MSE and standard deviation of RoMAE on the Pendulum dataset are reported in Table 5. RoMAE outperforms specialized Transformer based models such as ContiFormer [12], as well as state space models such as S5 [51].

## 5.5 Interpolation

We evaluate RoMAE on three interpolation tasks with increasing dimensionality and sampling irregularity. *(i) Spiral*: A 2D synthetic benchmark of 300 noisy Archimedean spirals as in Ref. [12]; *(ii) Synthetic*: The 50-step univariate task from Ref. [48] and *(iii) PhysioNet*: 48-hour ICU records containing 41 clinical variables [49]. For *(i)*, each spiral is discretized into 75 evenly-spaced time steps. To create irregular time-series data, time points are randomly selected from the first half of each spiral, which are used for interpolation. For *(ii)*, the interpolation task is between a random subsample including between 3 and 10 points per trajectory. For *(iii)*, we follow the 50% masking protocol

of [35]: half of the time, rows with at least one observation are hidden and must be reconstructed from context. We provide additional experimental details in the Appendix, respectively D.7, D.8, and D.9. All experiments are conducted to ensure a direct comparison with the results of the Transformer [58], LatentODE [9], and ContiFormer [12] for *(i)*, and HeTVAE for *(ii)* and *(iii)*. Across scales, our RoMAE-tiny configuration consistently scores competitively with respect to benchmarked MSE/RMSE, as seen in Table. 7. We lastly remark on RoMAE's ability to retain progressively higher frequency modes for interpolation with time-series data in Appendix B.4.

## 6   Discussion

Table 7 shows that one tiny/small RoMAE model, pre-trained once with a generic masked-autoencoder objective and *no* task-specific architectural tuning, matches or surpasses specialised baselines across three increasingly difficult interpolation datasets. On the 2D *Spiral* benchmark, we improve by an order of magnitude over the previous best result by ContiFormer, which we attribute to MAE tubelet-masking enforcing long-range reasoning. For the 50-step *Synthetic* task we obtain $0.233 \pm 0.007$ MSE, comparable to HeTVAE's $0.223 \pm 0.070$ but with markedly tighter standard deviation (five seeds). On the irregular, 41-channel *PhysioNet-2012* ICU data we achieve $0.570 \pm 0.014$ MSE, within half a standard deviation of HeTVAE's heteroscedastic decoder. We have verified by retraining HeTVAE on the PhysioNet interpolation task that it indeed excels on densely sampled variables, such as heart-rate, whereas RoMAE maintains more balanced interpolation across the sparsely observed channels; consequently, while the aggregate MSE over all the features is similar, RoMAE delivers a simpler, single-stage model without task-specific adaptations.

These results indicate that RoMAE pre-training can be universally beneficial for continuous time interpolation over irregular time-series data. The RoMAE architecture is able to scale from low-dimensional position time embeddings (50,1), to large (2880,41) points with extremely sparse observations differing across features, without refinement of the inherent architecture, suggesting that MAE-style models can serve as strong, task-agnostic baselines for continuous-time interpolation. RoMAE's classification results on datasets sizes ranging from a few hundreds to millions show how pre-training enables RoMAE to be both scalable and data-efficient.

Our tests on position reconstruction demonstrate that care must be taken when working with learned embeddings. Given how prominent the `[CLS]` token is when working with Transformer Encoders, our results are relevant for a multitude of models, including RoFormer [54].

**Limitations:** RoPE in RoMAE has some additional computational overhead if the positions are different with each forward pass, e.g., with any continuous irregular time-series. If positions stay constant, however, as in images, the overhead becomes negligible. For details on the additional compute incurred by continuous RoPE, see Appendix B.2. RoMAE is also not well suited for very long sequences, as it uses standard Attention which has $O(n^2)$ memory complexity with regards to sequence length. Lastly, RoMAE's ability to perform on extrapolation tasks is limited, as discussed in Sec. B.3.

**Broader and Societal Impact:** We believe that RoMAE's flexibility can extend representation learning to scientific fields where it may not have been easily adopted thus far, e.g., similar to what we showed with the ELAsTiCC Challenge. The potential for the misuse of RoMAE is similar to that of MAE and other foundational models.

## 7   Conclusion

This paper introduced RoMAE, an extension to the Masked Autoencoder architecture that allows it to accept multi-dimensional data with continuous positions as input. We investigated the theoretical implications of using relative positional embeddings for an MAE model, showing how it changes the difficulty of the masked pre-training task. We also showed how the model can use a learned `[CLS]` token to recover absolute position. Results across a large variety of modalities and tasks have shown that RoMAE achieves excellent performance in modalities requiring continuous position while maintaining the performance of MAE on images and audio. Future work will include building a more robust theoretical understanding of the implications of using RoPE. Because RoPE is compatible with various Attention implementations, one could also adapt RoMAE to use a linear Attention variant, which would allow it to work with much larger input sequences. Finally, we envisage the exploration of the many potential modalities that RoMAE could work with that were not tested here.

## Acknowledgments

RT and AS acknowledge funding from Next Generation EU, in the context of the National Recovery and Resilience Plan, Investment PE1 Project FAIR "Future Artificial Intelligence Research". This resource was co-financed by the Next Generation EU [DM 1555 del 11.10.22]. RT is partially supported by the Fondazione ICSC, Spoke 3 "Astrophysics and Cosmos Observations", Piano Nazionale di Ripresa e Resilienza Project ID CN00000013 "Italian Research Center on High-Performance Computing, Big Data and Quantum Computing" funded by MUR Missione 4 Componente 2 Investimento 1.4: Potenziamento strutture di ricerca e creazione di "campioni nazionali di R&S (M4C2-19)" - Next Generation EU (NGEU). We also acknowledge the use of computational resources provided by the Italian AREA Science Park supercomputing platform ORFEO in Trieste. GC is supported by the European Union's Horizon Europe research and innovation program under the Marie Sklodowska-Curie COFUND Postdoctoral Programme grant agreement No.101081355- SMASH and by the Republic of Slovenia and the European Union from the European Regional Development Fund. Disclaimer: Co-funded by the European Union. Views and opinions expressed are however those of the author(s) only and do not necessarily reflect those of the European Union or European Research Executive Agency. Neither the European Union nor the granting authority can be held responsible for them.

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

# Appendix A   Model Details

## A.1   Model Sizes

Table 8: All RoMAE model sizes.

| Parameter | tiny-shallow | tiny | small | base |
|---|---|---|---|---|
| $d_{\text{model}}$ | 180 | 180 | 432 | 720 |
| $N_{\text{head}}$ | 3 | 3 | 6 | 12 |
| Depth | 2 | 12 | 12 | 12 |
| Dim. feed-forward | 720 | 720 | 1728 | 2880 |
| Num. parameters | 0.782M | 4.67M | 26.9M | 74.7M |

We define a set of model sizes for RoMAE which are based on the original BERT [14] and ViT [15] model sizes, and are given in Table 8. The most important difference between RoMAE sizes and the sizes of other BERT-style models is in the $d_{\text{model}}$ parameter: RoMAE adopts a different dimensionality because of the constraints regarding Axial RoPE described in Section 3.1. Specifically, we choose $d_{\text{model}}$ such that the same model dimensionality works up to 3 positional (axial) dimensions.

**Decoder Size:** Although we vary the size of the RoMAE Encoder throughout various training runs, we always use RoMAE tiny-shallow as the decoder size when pre-training. Our choice is motivated by MAE [23], which also uses a small and shallow decoder.

## A.2   Architectural, Normalization, and Regularization Details

Here we describe the components of RoMAE in more detail, the normalization techniques we use during various training runs and technologies we use to speed up training.

**RMSNorm:** RoMAE uses RMSNorm [66], which is defined as follows:

$$\bar{a}_i = \frac{a_i}{\text{RMS}(\mathbf{a})} g_i, \qquad \text{RMS}(\mathbf{a}) = \sqrt{\epsilon + \frac{1}{d_{\text{model}}} \sum_{i=1}^{d_{\text{model}}} a_i^2} \tag{3}$$

where $\mathbf{a}$ is a sequence of embeddings, $g_i$ is a learned parameter that rescales each $a_i \in \mathbf{a}$, and $\epsilon$ is a small value added for numerical stability. Notably, RMSNorm does not centre the input $\mathbf{a}$ like LayerNorm [2] does. Re-centring has been shown not to be necessary and removing it saves compute.

**Patch Reconstruction:** After passing all [MASK] tokens through the RoMAE decoder, we pass the same reconstruction head over all [MASK] tokens to predict the original patch values. This head consists of an RMSNorm followed by a linear layer $W^{d_{\text{model}} \times n_p}$.

**Classification Head:** The classification head we use has the same structure as the patch reconstruction head, using an RMSNorm and a linear layer $W^{d_{\text{model}} \times n_{\text{classes}}}$. We place the head on top of the [CLS] token when it is available. Otherwise we take the mean of the output embeddings and place the head on top of this.

**Stochastic Depth:** In some runs we use stochastic depth [27], which is a form of dropout where whole layers are zeroed out. Specifically, each layer $l_m$ which has depth $m$, has a probability $\frac{\lambda m}{N_{\text{layers}}}$ to be zeroed out, where $\lambda$ is the probability of the final layer being zeroed out.

**Mixed Precision Training:** Although all final results are in full FP32 precision, we also tried mixed precision training through PyTorch Automatic Mixed Precision (AMP).[5] When training with AMP, some operations are conducted in a lower precision (either 16-bit brain floating-point (BF16) or 16-bit floating-point (FP16)) instead of the usual 32-bit floating point. This speeds the model up greatly, resulting in significantly less compute resources being used. In our experiments we found that RoMAE still converged well when using mixed precision.

**Dropout:** When using dropout [26], we apply it to the attention scores and to the MLP hidden layer with the same probability.

---

[5] `https://docs.pytorch.org/docs/stable/amp.html`

**Label smoothing:** To help reduce overfitting, label smoothing [56] prevents the model from becoming overconfident. This is done by changing the model target labels, reducing each correct class label from 1 to a confidence value $c$, and increasing all incorrect class labels from a value of 0 to a value of $(1 - c)/n_{\text{classes}}$.

## Appendix B  Additional Experiments and Discussion

### B.1  Additional Absolute Position Reconstruction Results

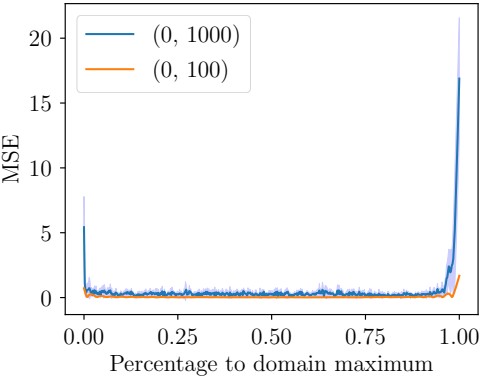

Figure 2: RoMAE position reconstruction MSE across two positional ranges.

Here we provide an additional experiment showing how RoMAE is able to to reconstruct absolute position across a wide range of values. To conduct the experiment, we pass only one token into RoMAE, giving it a random position drawn from a uniform distribution $\mathcal{U}_{[a,\,b]}$, training the model to predict this position. We also pass in the [CLS] token. The experiment is conducted over two domains; $(0, 100)$ and $(0, 1000)$. Hyperparameters and training details are discussed in Section D.5. After training, we evaluate how well the model performs across the range of values it was trained on. Results are plotted in Figure 2.

**Discussion:** We find that the model is generally able to reconstruct all position values within the two domains tested, except when the position is close to the edges of the domain. This effect occurs already when the domain is relatively small and worsens as it becomes larger. We also find that the model performs better overall on the smaller domain. These results provide empirical support for Proposition 4.2, and show how the model is able to learn to reconstruct absolute position across a large domain. That the loss grows as the position nears the edges of the domain shows that the model does not find solutions that generalize to out-of-distribution positions. These results also indicate that it may be beneficial to rescale positions to be within a smaller range.

### B.2  Compute Performance

Table 9: Relative speed of RoMAE when used with regular/irregular positions.

| Positional Embedding | Relative speed |
| --- | --- |
| Absolute (sin/cos) | 1 |
| RoPE (quantized) | 0.98 |
| RoPE (continuous) | 0.87 |

We evaluate the performance of RoMAE when using different positional encoding methods, specifically: absolute sin/cos [58], RoPE with integer (quantized) positions, and RoPE with continuous positions. The workload we test on uses 2 positional dimensions for RoPE and 2D image-like inputs to the model. Therefore, this experiment is representative of what one would encounter when using

data such as what we have in the Tiny ImageNet experiment (Section 5.2), or in the ELAsTiCC experiment (Section 5.4). The results, calculated on an NVIDIA 1650Ti GPU, are shown in Table 9.

Although the performance of regular quantized RoPE is not far from standard absolute positional embeddings, when switching to continuous RoPE the model is only 87% of the original speed. This is because we are unable to cache the RoPE frequencies between forward passes. With quantized position on the other hand, everything can be cached once before-hand and reused. While continuous RoPE incurs a notable performance penalty, it is not drastic. We note that other architectures specialized in irregular time-series also suffer from this issue, e.g., ContiFormer [11] is reported as being 6 times slower than the vanilla transformer. The performance of RoMAE could likely be improved through a more optimized RoPE implementation. Quantizing the positions could also address this issue in datasets where it is reasonable to do.

## B.3 Extrapolation

Being a BERT-style model, RoMAE is not well suited for extrapolation. During training the bidirectional encoder sees all tokens that lie inside the observed temporal window, therefore it never learns an inductive bias for causal ordering or forward progress in time, and struggles with out-of-distribution positions during inference. Recent work on causal Transformers, for example GPT-family models equipped with RoPE [52] or exponential relative embeddings [54, 53], shows that a strictly unidirectional attention pattern together with position embeddings that extrapolate to unseen indices can capture temporal trends far more effectively. We argue that despite this limitation, RoMAE has a place as a representation learning and interpolation framework, similarly to BERT in language.

## B.4 Retaining Frequencies

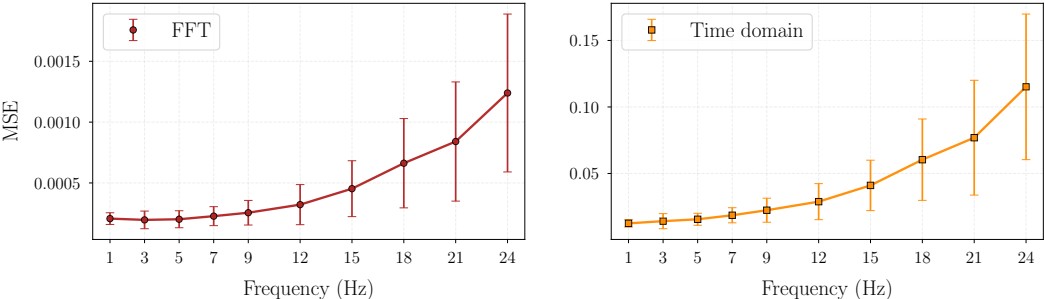

Figure 3: Average MSE obtained from the interpolation task using RoMAE-tiny for time-series with a single varying frequency component. **Left**: MSE computed on the Fast Fourier Transform (FFT). **Right**: MSE in the time domain. We generate 200 time-series per individual frequency, with 50 observed noisy points and 50 masked (interpolated) points, thus a limiting frequency of 25 according to Nyquist-Shannon sampling theorem. Error bars show the standard deviation of the MSE obtained for each individual frequency.

Here we investigate RoMAE's ability to retain high frequency modes in interpolation tasks. It is known, for example as shown in References [42, 64], that neural networks can exhibit a spectral bias, in that the networks preferentially learn low-frequency components before high-frequency details. Examining how RoMAE reconstructs patterns at different frequencies provides insight into whether the rotational encoding allows to capture fine-grained structure during interpolation, with implications for understanding the inductive biases introduced by this positional encoding scheme.

To empirically assess RoMAE's ability to reconstruct signals at different frequencies, we designed a controlled toy dataset of noisy sine waves. Each time series is defined over $t \in [0, 1]$ and generated as the sum of one or two frequency components (with equal probability), where integer frequencies are sampled uniformly from $f \in [0, 24]$ Hz and Gaussian noise $\varepsilon \sim \mathcal{N}(0, 0.01)$ is added. Each time-series has 100 data points, 50 of which are taken as input and 50 of which are masked for interpolation. We train RoMAE-tiny for 200 epochs on 10,000 examples. We then evaluate this model on (i) time-series with a single frequency mode as shown in Figure 3, (ii) time-series with two frequencies modes as shown in Figure 4. Using the 50 predicted (interpolated) points, we compute the

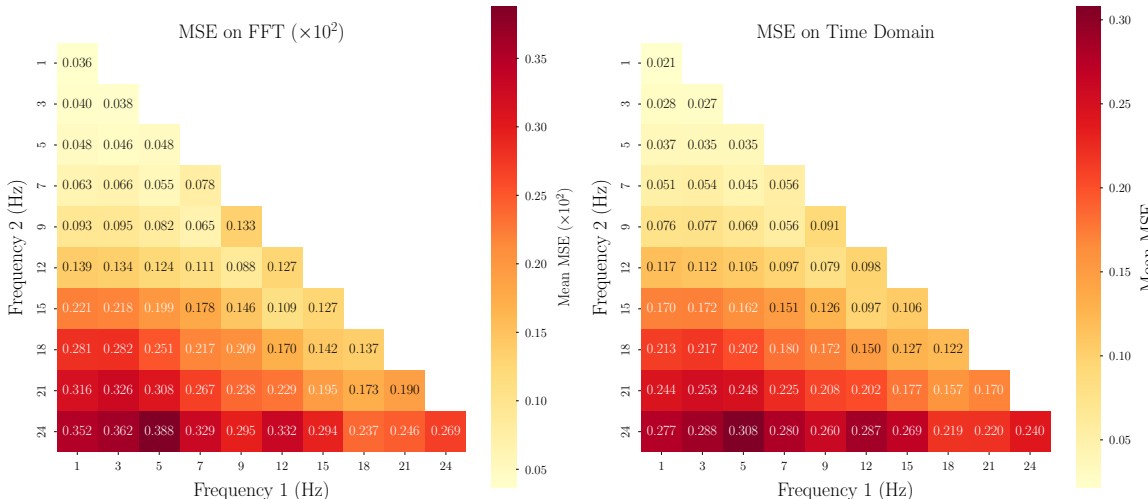

Figure 4: Same as Figure 3, but now for time-series with two frequency modes present in the signal. **Left**: MSE computed on the FFT. **Right**: MSE in the time domain. The time-series have 50 observed noisy points and 50 masked (interpolated) points.

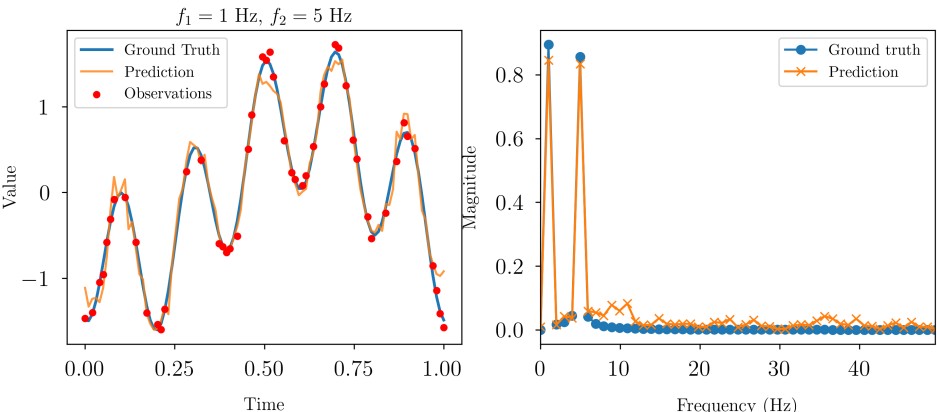

Figure 5: Illustrative realisation from the evaluation of RoMAE on a the bi-frequency time series. **Left**: Interpolation in the time domain for a composite sinusoidal signal with base frequencies 1 and 5 Hz. **Right**: FFT of the ground truth and predicted waveform.

MSE in both the time-domain and Fourier domain. We plot a sample prediction from the evaluation in Figure 5. This analysis was conducted with a mean signal to noise ratio (SNR) of 43.1 over the entire training set. We acknowledge that the distribution of the MSE will be affected by an increasing SNR, resulting in less sensitivity to higher frequency modes.

As expected, we observe a general degradation of the reconstruction for higher frequencies, with an approximately linear trend in error for frequency above 9Hz. For the case of two modes, we observe the same overall trend with a slight preference toward two higher frequency modes, as opposed to one low and one high mode. This is due to the sampling rate of the observations and the fact that the FFT for two higher frequencies has a uni-modal power spectrum, yielding slightly better reconstruction. Lastly, we have checked that the above observations are maintained for non-sinusoidal signals. We repeated the analysis using non-sinusoidal periodic functions, specifically a square wave and a cycloid. We observed similar behaviour for the retention of high frequencies as with the sinusoidal experiment.

# Appendix C    Proofs

**Notation:** Here we define additional notation, on top of what is presented in Section 3.1. We define the block-diagonal rotation matrix $\mathbf{R}$ which contains the 2D rotation matrices corresponding to all $\theta_i$'s:

$$\mathbf{\Theta}_i = \begin{pmatrix} \cos\,\theta_i & -\sin\,\theta_i \\ \sin\,\theta_i & \cos\,\theta_i \end{pmatrix}, \qquad \mathbf{R} = \begin{pmatrix} \Theta_1 & 0 & \cdots & 0 \\ 0 & \Theta_2 & \cdots & 0 \\ \vdots & \vdots & \ddots & \vdots \\ 0 & 0 & \cdots & \Theta_{d_{\mathrm{model}}/2} \end{pmatrix}. \tag{4}$$

When applying RoPE, $\mathbf{R}$ is exponentiated by position $m$, then multiplied by $x_m$. E.g.: $\mathbf{R}^m x_m$.

**Definition C.1** ([CLS] token). The learned [CLS] token is a vector $x_{\mathrm{CLS}} \in \mathbb{R}^{d_{\mathrm{model}}}$ consisting of learnable parameters, that is appended to the start of sequence $\mathbf{z}$ as described in Section 4. The position of $x_{\mathrm{CLS}}$ is always zero.

## C.1    Reconstructing Absolute Position Using the [CLS] Token

We now prove Proposition 4.2 by construction. The proof is based on the proof by Barbero et al. [4], showing that RoPE can be maximized for any relative distance $r \in \mathbb{Z}$. Here we generalize this result to a continuous position $r \in \mathbb{Q}$.

**Proof:** Consider a distance $r \in \mathbb{Q}^+ \subset \mathbb{R}$, a query $\mathbf{q} = \boldsymbol{\psi}$ that is non-zero by assumption and a key corresponding to the [CLS] token as described in Definition C.1 such that $\mathbf{k} = \mathbf{R}^r \boldsymbol{\psi}$. Assume that the query is at position $j \in \mathbb{Q}^+$. We compute the dot product between rotated $\mathbf{q}$ and $\mathbf{k}$:

$$\left(\mathbf{R}^j \mathbf{q}\right)^\top \left(\mathbf{R}^0 \mathbf{k}\right) = \mathbf{q}^\top \mathbf{R}^{-j} \mathbf{k} = \boldsymbol{\psi}^\top \mathbf{R}^{-j+r} \boldsymbol{\psi} \tag{5}$$

We now write this as the sum of dot products between the $\Theta_i$'s and each 2D subspace $\boldsymbol{\psi}^{(i)}$:

$$= \sum_{i=1}^{d_{\mathrm{model}}/2} \left(\boldsymbol{\psi}^{(i)}\right)^\top \Theta_i^{-j+r} \boldsymbol{\psi}^{(i)} \tag{6}$$

$$= \sum_{i=1}^{d_{\mathrm{model}}/2} \left\|\boldsymbol{\psi}^{(i)}\right\|^2 \cos\left((-j+r)\theta_i\right) \tag{7}$$

Because both $j$ and $r$ are in $\mathbb{Q}^+$, and $\theta_i$ is never a multiple of $\pi$ by definition, the unique maximum occurs when $j = r$. A similar proof applies when $r, j \in \mathbb{Q}^-$. $\qquad\square$

# Appendix D    Full Experimental Details and Hyperparameters

## D.1    Tiny Imagenet Experimental Setup

We present the unified Tiny ImageNet pre-training and fine-tuning hyperparameters in Table 10. All Tiny ImageNet pre-training and fine-tuning runs use the same hyperparameters. Although our final results use FP32, we also tested mixed-precision training and found that FP16 precision works with Tiny-ImageNet as opposed to our experiences on the ELAsTiCC dataset discussed in Section D.6.

When training, we normalize using the ImageNet mean and standard deviation. Each patch is individually normalized when calculating loss as is done in MAE [23]. During fine-tuning, we also use RandAugment [13] with 2 operations and a magnitude of 9. While the model would likely benefit from more epochs during the pre-training and fine-tuning stages, we consider this sufficient for the purposes of an ablation study.

Table 10: Tiny ImageNet unified pre-training and fine-tuning hyperparameters.

| Hyperparameter | Pre-Training | Fine-Tuning |
|---|---|---|
| Optimizer | AdamW | AdamW |
| AdamW betas | $\beta_1 = 0.9, \ \beta_2 = 0.95$ | $\beta_1 = 0.9, \ \beta_2 = 0.999$ |
| Weight Decay | 0.05 | 0.05 |
| Base LR | $2 \times 10^{-4}$ | $1 \times 10^{-3}$ |
| Batch size | 1024 | 1024 |
| Epochs | 200 | 15 |
| Gradient clip | 1 | 1 |
| Linear LR warmup steps | 2000 | 500 |
| LR schedule | Cosine | Cosine |
| Dropout | 0 | 0 |
| Stochastic depth $\lambda$ | 0 | 0 |
| Label smoothing $c$ | – | 0.9 |
| Precision | FP32 | FP32 |

## D.2 Audio Experimental Setup

In Table 11 we present the unified pre-training and fine-tuning hyperparameters for the audio representation learning and classification experiments, discussed in Section 5.3, in the main text.

When training, both on Audioset [20] and Librispeech [36], we normalize the data using the mean and standard deviation estimated on the entire set of spectrograms. Each patch is individually normalized before calculating the loss as is done in MAE [23].

Table 11: Audio Experiment unified pre-training and fine-tuning hyperparameters.

| Hyperparameter | Pre-Training | Fine-Tuning |
|---|---|---|
| Optimizer | AdamW | AdamW |
| AdamW betas | $\beta_1 = 0.9, \ \beta_2 = 0.95$ | $\beta_1 = 0.9, \ \beta_2 = 0.99$ |
| Weight Decay | 0.05 | 0.02 |
| Base LR | $5 \times 10^{-4}$ | $1 \times 10^{-3}$ |
| Batch size | 64 | 48 |
| Epochs | 150 | 50 |
| Gradient clip | 1 | 1 |
| Linear LR warmup steps | 1000 | 50 |
| LR schedule | Cosine | Cosine |
| Dropout | 0 | 0 |
| Stochastic depth $\lambda$ | 0 | 0 |
| Label smoothing $c$ | – | 0.8 |
| Precision | FP32 | FP32 |

## D.3 UEA Multivariate Time-series Experimental Setup

We use the same pre-training setting across all datasets, shown in Table 13. When fine-tuning, we find it necessary to have per-dataset hyperparameters. These are shown in Table 12. Because the datasets have varying sizes, resulting in vastly different numbers of training steps, we choose to scale the number of learning rate warmup steps as a percentage of total steps.

## D.4 Pendulum Dataset

When training on the Pendulum dataset, we use a custom model size shown in Table 14. The model size is chosen such that the MLP hidden dimension is equal to that of other models benchmarked on the dataset. To create the dataset, we follow the procedure from S5 [51], using their published code[6].

---

[6]`https://github.com/lindermanlab/S5/tree/pendulum`

Table 12: UEA Multivariate Time-series Archive fine-tuning hyperparameters. Values that are constant across all runs are reported only once.

| Hyperparameter | BM | CT | EP | HB | LSST |
|---|---|---|---|---|---|
| Optimizer | | | SGD | | |
| Momentum | | | 0.9 | | |
| Weight Decay | | | 0. | | |
| Base LR | $1 \times 10^{-2}$ | $8 \times 10^{-3}$ | $2 \times 10^{-3}$ | $2 \times 10^{-2}$ | $3 \times 10^{-2}$ |
| Batch size | 8 | 16 | 16 | 32 | 16 |
| Epochs | 50 | 100 | 150 | 30 | 15 |
| Gradient clip | 1 | 1 | 1 | 2 | 10 |
| Linear LR warmup percentage | | | 10% | | |
| LR schedule | | | Cosine | | |
| Dropout | 0 | 0 | 0.2 | 0 | 0.2 |
| Stochastic depth $\lambda$ | 0 | 0 | 0.2 | 0 | 0.2 |
| Label smoothing $c$ | 1 | 0.9 | 0.8 | 1 | 0.9 |
| Precision | | | FP32 | | |

Table 13: UEA Multivariate Time-series Archive unified pre-training hyperparameters.

| Hyperparameter | Value |
|---|---|
| Optimizer | AdamW |
| AdamW betas | $\beta_1 = 0.9, \ \beta_2 = 0.95$ |
| Weight Decay | 0.05 |
| Base LR | $3 \times 10^{-4}$ |
| Batch size | 64 |
| Epochs | 400 |
| Gradient clip | 1 |
| Linear LR warmup percentage | 0.1 |
| LR schedule | Cosine |
| Dropout | 0 |
| Stochastic depth $\lambda$ | 0 |
| Precision | FP32 |

Training RoMAE on the Pendulum dataset is generally very fast because of the small model size and the lack of pre-training.

### D.5 Absolute Position Reconstruction Hyperparameters

Here we provide full details for the experiments shown in Section 5.1 and Appendix B.1. These results can be seen in Table 16 and Table 17, respectively. Across both experiments we keep the model size equal to RoMAE-tiny as described in Table 8 except for $d_{\mathrm{model}}$ which we set to 960 for the experiment in Section B.1, and set according to the corresponding model size for the experiment in Section 5.1. All experiments in Section 5.1 use the same hyperparameters shown in Table 17. For both experiments we report the mean and standard deviation across 5 different seeds.

### D.6 ELAsTiCC Experimental Setup

Full pre-training and fine-tuning hyperparameters for Section 5.4 can be found in Table 18. When creating the train/test split we use the code and pre-processing provided in the ATAT [7] code release[7]. The alert mask removes points whose flux is either saturated or has a high estimated error. A saturated flux occurs when the flux is higher than the maximum amount that the instrument measuring it can record.

Because the input values in ELAsTiCC are nearly always larger than what one would find with images (e.g., of the order of 10-50 when standardized as opposed to between 0 and 1 with images),

---
[7]https://github.com/alercebroker/ATAT

Table 14: Pendulum dataset custom model size.

| Model Parameter | Value |
|---|---|
| $d_{\text{model}}$ | 60 |
| $N_{\text{head}}$ | 2 |
| Depth | 2 |
| Dim. feed-forward | 30 |
| Num. parameters | 37.4K |

Table 15: Pendulum dataset end-to-end training hyperparameters.

| Hyperparameter | Value |
|---|---|
| Optimizer | AdamW |
| AdamW betas | $\beta_1 = 0.9,\ \beta_2 = 0.999$ |
| Weight Decay | 0.01 |
| Base LR | $3 \times 10^{-4}$ |
| Batch size | 16 |
| Epochs | 50 |
| Gradient clip | 1 |
| Linear LR warmup steps | 1000 |
| LR schedule | Cosine |
| Dropout | 0 |
| Stochastic depth $\lambda$ | 0 |
| Precision | FP32 |

Table 16: End-to-end training hyperparameters for the absolute reconstruction range experiment (Section B.1). Values that are constant across all runs are reported only once.

| Hyperparameter | (0, 100) | (0, 1000) |
|---|---|---|
| Optimizer | SGD | |
| Momentum | 0.9 | |
| Weight Decay | 0. | |
| Base LR | $5 \times 10^{-6}$ | $5 \times 10^{-7}$ |
| Batch size | 64 | |
| Epochs | 10 | |
| Gradient clip | inf | |
| Linear LR warmup steps | 625 | |
| LR schedule | Cosine | |
| Dropout | 0 | |
| Stochastic depth $\lambda$ | 0 | |
| Precision | FP32 | |

Table 17: End-to-end training hyperparameters for the absolute reconstruction MSE experiment (Section 5.1).

| Hyperparameter | Value |
|---|---|
| Optimizer | AdamW |
| AdamW betas | $\beta_1 = 0.9,\ \beta_2 = 0.999$ |
| Weight Decay | 0.01 |
| Base LR | $5 \times 10^{-4}$ |
| Batch size | 64 |
| Epochs | 10 |
| Gradient clip | inf |
| Linear LR warmup steps | 625 |
| LR schedule | Cosine |
| Dropout | 0 |
| Stochastic depth $\lambda$ | 0 |
| Precision | FP32 |

we found it beneficial to increase the gradient clip threshold to 10 from the common value of 1. In order to handle the variable number of points per sample we utilize padding, applying a pad mask to the attention scores. Although our final model was trained using full FP32 precision, we tested RoMAE with both FP16 and BF16 for mixed precision training, and found that FP16 resulted in NaN

Table 18: ELAsTiCC full training hyperparameters.

| Hyperparameter | Pre-Training | Fine-Tuning |
|---|---|---|
| Optimizer | AdamW | AdamW |
| AdamW betas | $\beta_1 = 0.9,\ \beta_2 = 0.95$ | $\beta_1 = 0.9,\ \beta_2 = 0.999$ |
| Weight Decay | 0.05 | 0.05 |
| Base LR | $6.4 \times 10^{-3}$ | $8 \times 10^{-4}$ |
| Batch size | 16384 | 4096 |
| Epochs | 200 | 25 |
| Gradient clip | 10 | 10 |
| Linear LR warmup steps | 2000 | 2000 |
| LR schedule | Cosine | Cosine |
| Dropout | 0 | 0.2 |
| Stochastic depth $\lambda$ | 0 | 0.2 |
| Precision | FP32 | FP32 |

values. This is likely due to to the increased input range in ELAsTiCC interacting poorly with the reduced range of FP16. We found that BF16, with its larger range, worked well.

Each light curve in the ELAsTiCC dataset contains recordings of both the flux difference and variance. An example training sample is visualized in Figure 6, showing how each band has a different number of observations, each of which is at a different time than the others. To convert each point in the light curve to an embedding we use a patch size of (1, 2) for time and flux/variance respectively. Therefore, during pre-training the model predicts not only the masked flux difference values but also variance for each point, while time and band index are embedded using position.

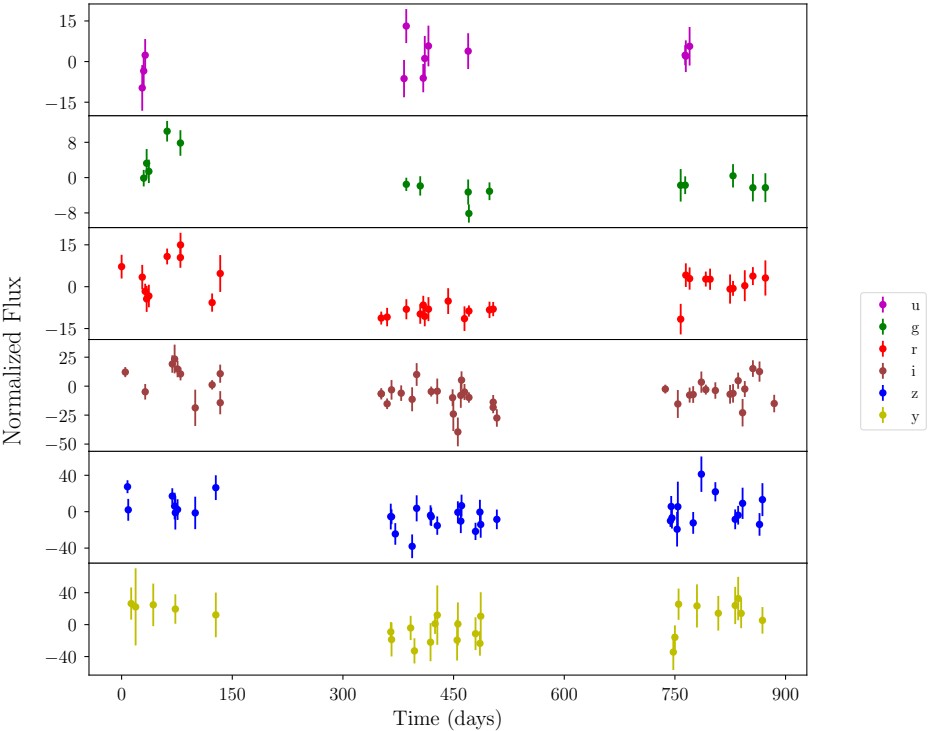

Figure 6: A training example from the ELAsTiCC dataset. The flux difference of each band has already been normalized.

Table 19: End-to-end training hyperparameters for the spirals dataset.

| Hyperparameter | Value |
|---|---|
| Model size | Tiny |
| Optimizer | AdamW |
| AdamW betas | $\beta_1 = 0.9,\ \beta_2 = 0.999$ |
| Weight Decay | 0.01 |
| Base LR | $3 \times 10^{-4}$ |
| Batch size | 32 |
| Epochs | 500 |
| Gradient clip | 1 |
| Linear LR warmup steps | 2000 |
| LR schedule | Cosine |
| Dropout | 0 |
| Stochastic depth $\lambda$ | 0 |
| Precision | FP32 |

## D.7 Spiral dataset

We construct a dataset of 300 spirals as per the prescription from Ref. [12], similarly allocating 200 for training and 100 for testing. Each spiral is randomly assigned to be either clockwise or counter-clockwise, with parameters drawn from normal distributions

$$a \sim \mathcal{N}(0, \alpha) \quad \text{and} \quad b \sim \mathcal{N}(0.3, \alpha),$$

where $\alpha = 0.02$. For the results presented in Table. 7 and for comparison with ContiFormer, we add Gaussian noise sampled from $\mathcal{N}(0, \beta)$ to the training samples, setting $\beta = 0.1$. The spirals were truncated at times corresponding to $6\pi$ in both cases, and only the parts of the spiral corresponding to the interpolation task carried out by Ref. [12] were used. Each spiral is discretized into 75 evenly spaced time steps. To create irregular time series data, 30 time points are randomly selected from the first half of each spiral, which are used for interpolation. We show the model hyperparameters used to generate the results in Table 19.

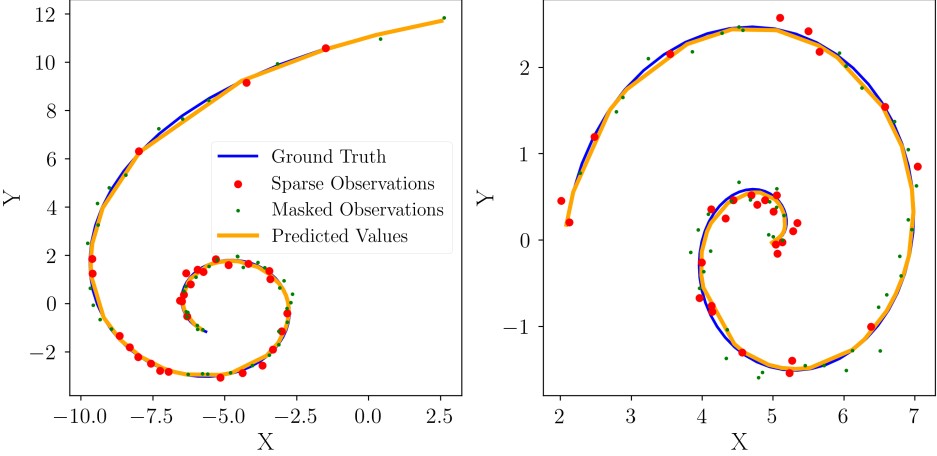

Figure 7: Two sample realisations of differing chirality from the test set of spirals. The green line is the ground truth trajectory. The Red points are the 75 stochastic inputs of which 45 are masked. The blue points are the interpolated predictions.

The $x$ and $y$ coordinates of the spirals are embedded using a patch size of (1,2) for time and x/y coordinates respectively. We train for 400 epochs with a learning rate of $10^{-3}$. Uncertainties presented in Table 7 represent the evaluation uncertainties after 10 trials with randomly seeded batches of 100 test spirals. The addition of the CLS token was observed to not significantly improve

performance. The code for generating the exact spirals used for this experiment, as well the details of the experimental setup for ContiFormer on their github [8].

## D.8  Synthetic dataset

Table 20: Synthetic dataset training hyperparameters

| Hyperparameter | Value |
| --- | --- |
| Model size | Tiny |
| Optimizer | AdamW |
| AdamW betas | $\beta_1 = 0.9, \ \beta_2 = 0.999$ |
| Weight Decay | 0.01 |
| Base LR | $1 \times 10^{-3}$ |
| Batch size | 8 |
| Epochs | 50 |
| Gradient clip | 1 |
| Linear LR warmup steps | 200 |
| LR schedule | Cosine |
| Dropout | 0 |
| Stochastic depth $\lambda$ | 0 |
| Precision | FP32 |

We evaluate RoMAE on a synthetic interpolation task introduced by Ref. [47], using the same code to generate the dataset[9]. The dataset comprises 2,000 univariate time series, each with 50 uniformly spaced time points in $[0, 1]$. With a patch size of $(1, 1)$, each individual point is converted to a token. Each trajectory is generated by sampling 10 latent variables $z_k \sim \mathcal{N}(0, 1)$ at reference times $r_k = 0.1 \cdot k$, and applying an RBF kernel smoother with bandwidth $\alpha = 120.0$ to interpolate values across the timeline. Gaussian noise $\mathcal{N}(0, 0.01)$ is added to simulate measurement error. To mimic irregular sampling, we randomly select between 3 and 10 observed points per trajectory. The dataset is split into $80\%$ training, $10\%$ validation, and $10\%$ test sets. Performance is assessed using mean squared error (MSE). We display results from some test samples for a number of observed points $n = 2, 10$ and 20 in Figure 8. The addition of the CLS token was observed to not significantly impact performance. We show the model hyperparameters used to generate the results in Table 20.

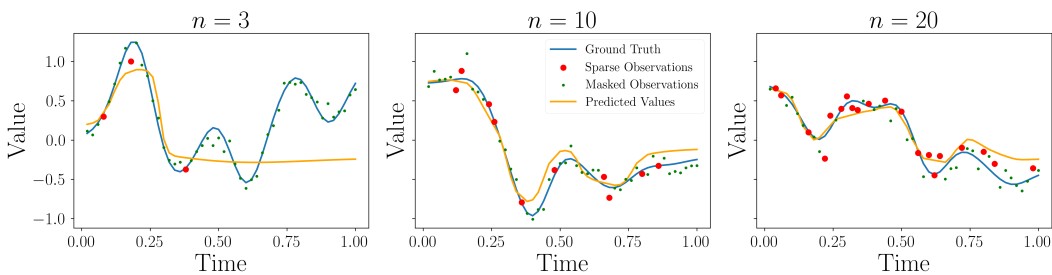

Figure 8: Samples from interpolation tests using $n = 3, 10$ and 20 observations.

## D.9  PhysioNet

We adopt the pre–processed release of the PHYSIONET/CinC 2012 Challenge [50], comprising multivariate clinical time–series collected during the 48h window following intensive–care–unit (ICU) admission. Static covariates (*Age*, *Gender*, *Height*, *ICU type*) occupy feature indices 0–3 and are always observed, whereas the remaining 37 channels are sparsely and irregularly sampled.

---

[8]https://github.com/microsoft/SeqML/tree/main/ContiFormer
[9]https://github.com/reml-lab/hetvae/blob/main/src/utils.py

In order to benchmark RoMAE we directly compare performance on the interpolation task using the same pre-processed version of the dataset produced by Ref. [35][10], which rounds the observation times to the nearest minute resulting in 2880 possible measurement times per time series. The data set includes 8000 instances that can be used for interpolation experiments. We additionally use the same experimental protocols which involve masking 50% of observed time points. Each multivariate record in the PHYSIONET 48 h clinical dataset is converted into a sequence of *scalar tokens* that RoMAE can process. Let $x_t^{(d)} \in \mathbb{R}$ denote the value of feature $d \in \{1, \ldots, 41\}$ measured at minute–resolution time step $t \in \{1, \ldots, T\}$ ($T \leq 2880$); let $m_t^{(d)} \in \{0, 1\}$ be the corresponding observation mask (1 = measured). We flatten the spatio–temporal grid into a one–dimensional token list $\{(x_n, p_n)\}_{n=1}^N$ with

$$x_n = x_t^{(d)}, \qquad p_n = \left[t/T, \ d\right]^\top, \qquad N = \sum_{t,d} 1.$$

The two–dimensional positional vector $p_n$ encodes (i) the *normalised time* $t/T \in [0, 1]$ and (ii) the *feature index* $d$, providing the $n_{\mathrm{pos}} = 2$ co-ordinates required by RoMAE. During training we stochastically subsample 50% of the observed tokens. The final input tensor hence has length $N$ for the values, and shape $(2, N)$ for the positions, and a Boolean mask of length $N$ indicating which tokens RoMAE must reconstruct, exactly matching the interpolation protocol of the HeTVAE benchmark.

We show the results of interpolation study in Table 7 where we compare to HetVAE [35], as well as 8 other models benchmarked in that study. We show the model hyperparameters used to generate the results in Table 21 along with the addition of the CLS token that was seen to improve the results. Lastly, official Physionet challenge can be found on their website [11].

Table 21: PhysioNet dataset training hyperparameters

| Hyperparameter | Value |
| --- | --- |
| Model size | Tiny |
| Optimizer | AdamW |
| AdamW betas | $\beta_1 = 0.9, \ \beta_2 = 0.999$ |
| Weight Decay | 0.01 |
| Base LR | $1 \times 10^{-4}$ |
| Batch size | 16 |
| Epochs | 200 |
| Gradient clip | 1 |
| Linear LR warmup steps | 100 |
| LR schedule | Cosine |
| Dropout | 0 |
| Stochastic depth $\lambda$ | 0 |
| Precision | FP32 |

---

[10]`https://github.com/reml-lab/hetvae`
[11]`https://physionet.org/content/challenge-2012/1.0.0/`

