# OpenReview forum: "Rotary Masked Autoencoders are Versatile Learners"
_NeurIPS.cc/2025/Conference — NeurIPS 2025 poster_

### Official Review · Reviewer_Wjq7 · 2025-06-22

**Clarity:** 2
**Significance:** 2
**Originality:** 2
**Rating:** 4
**Confidence:** 3

**Summary:**

This paper proposes RoMAE, which extends Masked Autoencoders to handle irregular time-series by adapting RoPE for continuous positions. The key contribution is Continuous Axial RoPE that encodes real-valued timestamps, enabling a single unified architecture to achieve competitive performance across irregular time-series classification, standard image tasks, and interpolation without specialized modifications, demonstrating that general foundation models can match domain-specific approaches when equipped with appropriate continuous positional encoding.

**Questions:**

1.	How exactly do the authors handle the continuous extension of RoPE when timestamps have vastly different or when dealing with multi-dimensional irregular data where different dimensions have different temporal resolutions?
2.	The proposed masking appears to be uniform across patches, but irregular time-series naturally have varying information density. Are there any experiments with adaptive masking strategies that account for the irregularity pattern?
3.	The experiments cover various datasets, but can it be characterized how RoMAE's performance varies with different types of irregularity patterns (e.g., completely random sampling vs. burst sampling vs. periodic gaps)?

**Ethical Concerns:**

["NO or VERY MINOR ethics concerns only"]

**Final Justification:**

The authors' rebuttal addressed my primary concerns by providing new empirical data on computational overhead and offering clear technical clarifications on handling multi-dimensional irregular data. Given the thoroughness of the rebuttal and the new information provided, I raise my score from "3: Borderline reject" to "4: Borderline accept".

**Limitations:**

1.	The paper lacks discussion of how continuous RoPE approximations in practice might deviate from their theoretical guarantees, especially for extreme timestamp ranges.
2.	Limited analysis of how sensitive the method is to key hyperparameters (masking ratio, patch size, p-value for truncated RoPE) across different irregularity patterns.

**Paper Formatting Concerns:**

No major formatting issues.

**Quality:**

2

**Strengths And Weaknesses:**

Strengths:

1.	The authors provide formal analysis of position reconstruction capabilities and empirically validate these claims, offering both theoretical understanding and practical guidance on when to use [CLS] tokens for absolute vs. relative positioning.
2.	The paper demonstrates that RoMAE achieves competitive or superior performance on irregular time-series, standard vision tasks, and interpolation benchmarks, providing convincing evidence that a unified architecture can match domain-specific methods.
3.	The proposed method addresses the important challenge of handling irregular sampling in time-series data using standard Transformer components, eliminating the need for specialized architectures.

Weaknesses:

1.	While the application is novel, the main technical innovation—using RoPE with continuous values—is relatively straightforward since RoPE naturally accepts real-valued inputs, and the theoretical analysis mostly confirms expected behavior rather than revealing innovative insights.
2.	The approach is restricted to D ≤ 3 dimensions due to exponential token growth, and while the authors mention computational overhead for varying positions, they provide insufficient analysis of the practical computational costs compared to specialized time-series methods.
3.	The experimental evaluation lacks comparison with recent large-scale time-series models like Time-LLM or other foundation models specifically designed for time series data.

---

> ### Author Rebuttal · Authors · 2025-07-31
>
> We thank the reviewer for their useful feedback. We would like to firstly address some misunderstandings that were conveyed by the reviewer when specifying perceived limitations of our paper.
>
>
> ## Limitations
>
> **1.**
> See Appendix A.5 (Figure 2): We have explicitly investigated how RoMAE’s position reconstruction error changes depending on the range of the positions. This identifies that the error does in fact increase as the range of the positions increase. We will add an additional (more extreme) case (going up to ~10^6) to the plot as well as some additional discussion of this for the camera ready version.
>
> **2.**
> During our testing of the model we evaluated the relationship between irregular absolute position reconstruction and p-RoPE p value and found that using a value of 0.75 did not have any adverse effects on the reconstruction error. We agree that investigating the effect of irregularity on hyperparameters is an important consideration and will add some additional discussion regarding this to the Appendix.
>
>
> Turning now to the questions:
> ## Question 1
> Regarding different temporal resolutions and/or ranges: this is a very interesting point, which might be very relevant in some applications (e.g. astrophysics with different observational facilities/types, healthcare monitoring,...), thank you for pointing this out. There are different ways to handle such situations with RoMAE. First, it is possible to use RoMAE as is, since the position embedding through RoPE contains a large range of frequencies: the model then has the ability to choose to focus on relevant frequencies depending on the time-scale it is interested in, e.g., lower frequencies tell the model more about long time-scales (see for instancem  Barbero et al. 2024, “Round and round we go! What makes Rotary Positional Encodings useful?”).
>
>
> However, this “out-of-the-box” approach might suffer from limitations for extremely large positional ranges (potential numerical issues) and/or in the case of temporal imbalance between different data modalities. Another option is thus to use separate positional dimensions to encode different temporal resolutions (especially relevant in the case of different modalities observed at different time resolutions, as suggested by the reviewer). In this case, one could store, for example, seconds in one positional dimension, and years in another.
>
>
> ## Question 2
>
> Regarding adaptive masking, our current experiments do not use dataset-specific masking strategies as we are interested in providing a general baseline on which any adaptive masking strategy would only improve upon. We highlight that RoMAE is compatible with any masking strategy the user may want. For example, RoMAE would likely benefit from the tube masking strategy proposed by VideoMAE (arxiv:2203.12602) when being trained on video. Additionally, when considering interpolation, utilizing domain knowledge to construct a mask that is representative of actual missing data in the particular dataset being studied would also likely provide an improvement over a regular uniform mask. We will update the discussion for the camera ready version to clarify these points.
>
>
> ## Question 3
> Regarding analyzing the sensitivity/robustness of the performance with respect to the irregularity pattern (or degree): this is an interesting point which was also brought up by Reviewer 3 (joLj).   Indeed, this could be helpful for the community to understand the robustness of RoMAE to the degree/pattern of irregularity. A proper, thorough analysis would require designing a (synthetic) experiment where we control the irregularity and monitor the performance of RoMAE and other baselines. We note that to our knowledge this is not commonly done in the literature, and perhaps this experiment is better reserved for future work.
>
> More qualitatively, some of the current illustrations in the manuscript showcase that the datasets we used have varying degrees and patterns of irregularity that don’t seem to crucially impact RoMAE; e.g., in Fig 4 in the Appendix, a typical sample from ELAsTiCC, with large time-windows missing and –semi-periodic gaps–, and more subtle different sampling in each modalities, while the pendulum is somewhat more uniform while still irregular (see Fig 3 in Appendix); Section C7 in the Appendix also provides an illustration of the interpolation results at test-time with varying degree of sparsity (3, 10, 20 points), when trained on sequences with 3-10 datapoints.
>
>
> ## Considerations regarding dimensionality
> We would like to address some of the weaknesses brought up by the reviewer, notably regarding the claim that the approach is limited to D < 4 dimensions due to exponential token-growth. We want to clarify here that this only becomes an issue when dealing with many *irregular* positional dimensions. We have additionally demonstrated via use of the ELaSTICC dataset, how RoMAE can be used with 6 irregular dimensions. The usage of MAE in conjunction with a high masking ratio (in places where this is applicable) also helps alleviate numerical issues with token-growth. We also reiterate that, as mentioned in the text, RoMAE is compatible with linear attention variants, emphasizing its versatility.
>
> With regard to the comment about computational overhead (which has also been made by the other reviewers), we agree that it is important to measure this. Due to the limited time for this rebuttal, we have performed a quick test to estimate the  relative computational cost for a typical image (224x224) workload. The results are as follows:
> | Model | Relative speed (Baseline = MAE)|
> |-------------|-------------:|
> | MAE  | 1 |
> | RoMAE (regular positions)  | 0.98 |
> | RoMAE (irregular positions)  | 0.87 |
>
> We also invite the reviewer to look at our response to  Rev. 1 Q1 for more details and considerations on irregular dimensions with RoMAE .
>
> ## Comparisons to LLM’s for time-series
> Similar points were also raised by Reviewer’s 2 (mZNG) and 3 (joLj).
>
> When considering comparisons to autoregressive models such Time-LLM (Jin et al, 2024) and TimesFM (Google Research, 2024), that focus on the forecasting task, not on *masked-reconstruction* for representation learning and interpolation. RoMAE’s use of continuous RoPE within the MAE framework uniquely enables interpolation, a feature that is lacking from Time-LLM or TimesFM. Thus, their design goals and evaluation protocols differ fundamentally from ours: We aim for a unified encoder that can be fine-tuned or probed across downstream tasks (classification, interpolation), rather than a forecasting or point-forecasting model.

---

> > ### Comment · Reviewer_Wjq7 · 2025-08-06
> > **Thank you and a follow-up question**
> >
> > Thank you for your detailed rebuttal, which addressed my primary concerns by providing new empirical data on computational overhead and offering clear technical clarifications on handling multi-dimensional irregular data. Considering the thoroughness of the rebuttal and the new information provided, I am considering to raise my score.
> >
> > Referring to the rebuttal itself, there is still a follow-up question listed as follows.
> >
> > 1. The provided speed comparison is for a "typical image workload." Could you clarify if a similar overhead is expected for the irregular time-series tasks, such as the ELASTICC dataset, where the sequence lengths and dimensions are quite different from an image?

---

> > > ### Author Response · Authors · 2025-08-07
> > >
> > > Thank you very much for your feedback, as well as considering a score increase.
> > >
> > > To clear up your question regarding overhead on ELAsTiCC: The relative overhead will be the same, since both ELAsTiCC and our synthetic image dataset utilize 2 positional dimensions.

---

> > > > ### Comment · Reviewer_Wjq7 · 2025-08-09
> > > >
> > > > Thank you for the follow-up clarification.

---

### Official Review · Reviewer_joLj · 2025-07-01

**Clarity:** 3
**Significance:** 3
**Originality:** 4
**Rating:** 4
**Confidence:** 4

**Summary:**

The paper proposes a variation of rotary positional embeddings (RoPE), which is applied to the MAE architecture. On various benchmark tests, the proposed method shows robust performance.

**Questions:**

- For sparse cases, the patch size might matter as they can sometimes cover a too large area to have a meaningful number of samples, or a too small patch could result in too local representations. How does the proposed method handle such trade-off?

**Ethical Concerns:**

["NO or VERY MINOR ethics concerns only"]

**Final Justification:**

Although the method is a combination of existing ideas, the paper presents meaningful innovations.

**Limitations:**

yes

**Quality:**

3

**Strengths And Weaknesses:**

Strengths

- The RoPE method is nicely harmonized into the MAE framework.
- By using Axial RoPE on continuous values and by using patch-level input that comes with real-valued position, it implements the RoPE version that works on irregularly sampled data.
- It shows strong performance on Irregular classification tasks (e.g., ELAsTiCC, UEA datasets), interpolation tasks (e.g., PhysioNet, Synthetic), and irregular regression (Pendulum).

Weaknesses
- It has been mentioned that models like Latent ODE and HeTVAE can compete with their ability to use continuous time representation. It would be more insightful if these models are analyzed with computational costs.
- The proposed method shows strong performance on the interpolation task, but extrapolation beyond the scope of the observed data is not well studied.
- While the experiments were done on some irregular datasets, it is not thoroughly discussed how varying degrees of irregularity affects the performance. Very sparse data is not rare in machine learning applications.
- The proposed method appears to be somewhat incremental as it's based on the existing Axial RoPE method and patchfication.

---

> ### Author Rebuttal · Authors · 2025-07-31
>
> We thank the reviewer for their useful feedback. To answer their question on the patch size:
> Regarding the case of patches that are too small resulting in local representations: because of our use of bidirectional attention, every single embedding will have access to the full “information” globally, even if each original patch was very local. At the output of the model, there are no embeddings with only local information. However, the possible limitation here is that this could lead to many tokens. If this is the case, then one can follow what RoFormer (original RoPE implementation) did and use a linear attention variant. We would also like to add that the MAE framework in conjunction with a high masking ratio can also alleviate this issue.
>
> ## Computational cost vs. Latent ODE / HeTVAE
> We feel this is an excellent suggestion. We are currently working on such comparisons for the revised version of the manuscript. Due to the limited time for this rebuttal, we have performed a quick test demonstrating our relative speedup for a typical image (224x224) workload. The results are as follows:
> | Model | Relative speed (Baseline = MAE)|
> |-------------|-------------:|
> | MAE  | 1 |
> | RoMAE (regular positions)  | 0.98  |
> | RoMAE (irregular positions)  | 0.87  |
> We will add further comparisons to, e.g., Latent ODE or HeTVAE for the final version of the manuscript.
>
> ## Extrapolation
> As a BERT-style architecture, RoMAE is designed for interpolation and representation learning. RoMAE is, by design, not a GPT-like model which is tailor-fit for next token prediction (extrapolation), a limitation we will state explicitly in Sec 6 and the Abstract to avoid misleading claims or expectations.
>
> The limited extrapolation capacity we observe is not unique to RoMAE. Zeng et al. (2023) (arxiv:2205.13504) report that vanilla Transformer/GPT-style forecasters are outperformed by a one-layer linear baseline on nine long-horizon datasets, attributing the gap to poor temporal-length extrapolation. Likewise, Chen et al. (2024) (arxiv:2402.10635) note that discrete-time Transformers ‘have limitations in generalizing to continuous-time data paradigms, motivating their ContiFormer variant. Similar constraints are reported with GPT models for NLP utilizing regular RoPE (see ALiBi arxiv:2108.12409). Combining RoMAE with a causal decoder, for example, is a promising future work.
>
> ## Sensitivity to the degree of irregularity
> This is an interesting point which was also brought up by Reviewer Wjq7 (4). Indeed, this proposed study could be helpful for the community to understand the robustness of RoMAE to the degree/pattern of irregularity. However, a proper, thorough analysis would require designing a (synthetic) experiment where we control the irregularity and monitor the performance of RoMAE and other baselines. Moreover, we note that, to our knowledge, this type of investigation is not commonly done in the literature, and is perhaps better reserved for future work.
>
> More qualitatively however, some of the current illustrations in the manuscript showcase that the datasets we used have varying degrees and patterns of irregularity that don’t seem to crucially impact RoMAE; e.g., in Fig 4 in the Appendix, a typical sample from ELAsTiCC, with large time-windows missing and –semi-periodic gaps–, and more subtle different sampling in each modality, while the pendulum dataset is somewhat more uniform. Section A.8.4  in the Appendix also provides an illustration of the interpolation results at test-time with varying degrees of sparsity (3, 10, 20 points), when trained on sequences with 3-10 datapoints.
>
> ## The method is “incremental”
> While our contribution with RoMAE might come across as incremental (combining existing techniques), we want to point out that it is nonetheless novel (as this combination was not investigated before) and relevant given the performances shown. Additionally, our paper clarifies that RoMAE is more than a simple RoPE swap: we (i) extend RoPE to multi-axis, real-valued “continuous Axial RoPE” (ii) provide the first theoretical bound on when a [CLS] token suffices for absolute-position recovery; (iii) demonstrate a single encoder that, without domain-specific heads or pre-training, reconstructs images, audio, and highly irregular time-series across many tasks.

---

> > ### Comment · Reviewer_joLj · 2025-08-05
> > **Thank you**
> >
> > I appreciate the authors’ detailed rebuttal. It clarified some of my concerns on the patch size trade-offs in sparse scenarios and computational efficiency comparisons with continuous-time models (Latent ODE, HeTVAE). I think the computational comparisons with Latent ODE and HeTVAE still remain preliminary. I partly agree that the paper presents meaningful innovation despite the nature of a combination of existing ideas. I increased the originality score accordingly.

---

> > > ### Author Response · Authors · 2025-08-07
> > >
> > > Thank you very much. We appreciate your reply.

---

### Official Review · Reviewer_mZNG · 2025-07-02

**Clarity:** 3
**Significance:** 2
**Originality:** 2
**Rating:** 3
**Confidence:** 4

**Summary:**

This paper is focused on extending masked autoencoders (MAE) model for handling time-series data, but where the data are sampled at irregular time points. One way to do so would be to integrate position encodings/embeddings. This paper shows that incorporating rotary positional embeddings for continuous position encoding is a good idea. So, adapting and showing the potential value of RoPE to encode continuous temporal positions in MAE is the main contribution of this work. The experiments do show proof of feasibility in that in that on the datasets tested, the model achieves competitive performance without requiring specialized architectural modifications. There is some merit in showing that one model can handle both irregular temporal measurements and standard MAE use-cases.

**Questions:**

1. what were the core factors influencing the choice of baselines and datasets? Why were Mamba-based models and other more challenging datasets (where similar to what's described in the paper, irregular sampled data can be derived) not used?

2. The delta with S5 is small. Is there enough justification (due to significant benefits in other regime) to choose masked autoencoders for this task? There is at least some results questioning whether time-series data are a good fit for transformer based model.

3. How pathological are the irregular sampled data that doing a standard imputation as a pre-processing, they cannot be fed directly to Chronos or TimesFM?

**Ethical Concerns:**

["NO or VERY MINOR ethics concerns only"]

**Final Justification:**

I appreciate the feedback from the authors. I find the experimental evidence valuable overall. The scope of the problem and the novelty of the technical contribution remain somewhat moderate.

**Limitations:**

yes

**Quality:**

2

**Strengths And Weaknesses:**

strengths:

+As indicated in the summary, I find value in the experiments that a single model can achieve reasonable performance across multiple modalities (irregular time series, images, audio) without domain-specific modifications.

+The use of ROPE, while not very novel, addresses the need of applying MAE models on irregular temporal data. The theoretical analysis does not provide any new insight, but is still OK.

+As a proof of feasibility, I think that the experimental evaluation is reasonable. A few diverse tasks are covered and the model performs well across these applications.

+The approach likely maintains efficiency advantages of MAE although this is not really discussed in deep detail in the paper.

weakness:

-An inability to handle extrapolation is more limiting than the paper acknowledges. Forecasting is one of the core use cases of time-series models and a model that is natively inapplicable to this setting will only be useful in interpolation and representation learning. While this does not mean that such a model is not useful at all, but it does question the claim about versatility highlighted in the paper's title.

- The core technical piece is relatively incremental, substituting RoPE for standard positional encoding in MAE. The paper shows that this works - and so there is value in it -- but it does diminish the significance of the technical contribution.

- Experimental evaluations leave out recent Mamba-based approaches and focus on UEA (why MIMIC, UCR etc were left out is not clear). In terms of positioning, I would have expected the paper to make a stronger case for why state space models are not inherently better suited -- especially when the sequence lengths become much longer. This issue is also relevant given that the paper uses a sequence flattening approach which can make scalability an even bigger issue.

---

> ### Author Rebuttal · Authors · 2025-07-31
>
> We thank Reviewer 2 for their insightful response.
>
> ## Question 1
> We selected datasets that are both widely used for irregular multi-variate time-series benchmarks (UEA, PhysioNet) and directly comparable to prior MAE-style work (e.g., the synthetic interpolation dataset from HeTVAE). We could not run experiments on MIMIC due to its restricted credentialed access requirements.
>
> Furthermore, ELAsTiCC and UEA already provide a wide diversity in the datasets, with ELAsTiCC being significantly challenging for instance. We agree that adding evaluations on UCR would add value to the paper’s results, and are willing to conduct additional experiments on the UCR datasets.
>
> Regarding Mamba: its architecture is designed for causal, autoregressive generation, which is very different from our bidirectional approach. More simply, it aims to compete with autoregressive LLMs and not BERT-like models such as RoMAE. Instead, we found it more prudent to compare RoMAE to S5 (a state-space model known to excel at long sequences) to represent the strongest SSM baseline. This keeps our comparison “apples-to-apples”.
>
> ## Question 2
> It is true that on some tasks the raw accuracy gain over S5 is modest (and for one dataset S5 performs better); however, we want to point out that it is not always so (see e.g. results for EP, CT, in Table 4 of the manuscript). Ultimately, our goal is to demonstrate that RoMAE **can match** state-of-the-art SSMs on irregular time-series tasks without task-specific architectural tweaks, utilizing our smallest RoMAE architecture RoMAE-tiny. We also feel that our better results compared to other transformer-based models such as Contiformer (another masked-reconstruction model) exacerbate this fact. Lastly, bidirectional attention naturally captures global context for the interpolation task, a capability that causal models lack, making MAE‐style training particularly well suited for representation learning on sparse time series.
>
>
> ## Question 3
> In several cases, doing imputation (or interpolation) on the time series would be extremely difficult (and a problem in its own right). See for instance one example of an ELAsTiCC multi-variate time-series in Fig 4 in the Appendix which shows large windows of missing values; in addition to the windows of missing data to interpolate, in the specific case of ELAsTiCC (but likely relevant in other applications), the diversity of objects in the dataset makes it complicated to choose an interpolation/imputation strategy to model “well” all the time-series in the first place. Many researchers have expressed interest in approaches like RoMAE solely for its interpolation capabilities. The additional caveat of performing imputation as a pre-processing step is that if the imputation/interpolation is somewhat inaccurate or poor, this will inevitably impact the models used down the line (say e.g. as suggested Chronos). Avoiding this problem altogether seems like a benefit, at least in such cases where imputation would be non-trivial and potentially detrimental.
>
> When considering comparisons to autoregressive models such as Chronos or TimesFM, it’s important to note that they focus on forecasting on regularly sampled data, not on *masked-reconstruction* for representation learning and interpolation. RoMAE’s use of continuous RoPE within the MAE framework uniquely enables interpolation, a feature that is lacking from Chronos or TimesFM. Thus, their design goals and evaluation protocols differ fundamentally from ours: We aim for a unified encoder that can be fine-tuned or probed across downstream tasks (classification, interpolation), rather than a forecasting or point-forecasting model.
>
>
> Below, we additionally address some points the reviewer brought up:
> ### Extrapolation
> As a BERT-style architecture, RoMAE is designed for interpolation and representation learning. RoMAE is, by design, not a GPT which is tailor-fit for next token prediction, a limitation we will now state explicitly in Sec 6 and the Abstract to avoid misleading claims or expectations.
> The limited extrapolation capacity we observe is not unique to RoMAE. Zeng et al. (2023) (arxiv:2205.13504) report that vanilla Transformer/GPT-style forecasters are outperformed by a one-layer linear baseline on nine long-horizon datasets, attributing the gap to poor temporal-length extrapolation. Likewise, Chen et al. (2024) (arxiv:2402.10635) note that discrete-time Transformers ‘have limitations in generalizing to continuous-time data paradigms, motivating their ContiFormer variant. Similar constraints are reported with GPT models for NLP utilizing regular RoPE (see ALiBi arxiv:2108.12409). Combining RoMAE with a causal decoder, for example, is promising future work.
>
> ###  “Incremental” technical novelty
> While our contribution with RoMAE might come across as incremental (combining existing techniques), we want to point out that it is nonetheless novel (as this combination was not investigated before) and relevant given the performances shown. Additionally, our paper clarifies that RoMAE is more than a simple RoPE swap: we (i) extend RoPE to multi-axis, real-valued “continuous Axial RoPE” (ii) provide the first theoretical results on when a [CLS] token suffices for absolute-position recovery, using these theoretical results to justify the integrity of our approach; (iii) demonstrate a single encoder that, without domain-specific heads or pre-training, reconstructs images, audio, and highly irregular time-series across many tasks.

---

> > ### Comment · Reviewer_mZNG · 2025-08-03
> >
> > I appreciate your answers to the questions/comments in my review.
> >
> > I find your answer to Q2 quite reasonable. I was not criticizing the paper for under-performing some baselines models, which I think is OK. I was seeking clarification, which you have now provided. thank you.
> >
> > I need to refresh my memory regarding the other comments/questions in my review and will write after I do that and reading the other reviews more carefully.

---

> > > ### Comment · Reviewer_mZNG · 2025-08-06
> > >
> > > I have now carefully gone through the paper again, the other reviews as well as your response. Sorry for the delay.
> > >
> > > I think that my current assessment is accurate for the following reason.
> > >
> > > The paper boils down to three existing pieces: (1) MAE, (2) RoPE position encoding (for continuous) and (3) Axial RoPE for multi-dimensions.
> > >
> > > The analysis regarding  absolute position recovery is a minor extension of existing work. By itself, this is fine but doesn't convey much more.
> > >
> > > Showing that components above work well together does have empirical value, but the novelty is still limited in my view. The experiments validate that substituting the encoding with RoPE, and applying it to continuous positions (which was supported), produces a working system for irregular time-series. I still don't buy the bit about versatility but that's a minor nit.

---

> > > > ### Author Response · Authors · 2025-08-07
> > > >
> > > > We appreciate your views, although we do wish to challenge your point regarding absolute position recovery.
> > > >
> > > > Our theoretical work on this matter is not only required to justify our proposed method, but it also provides a novel mechanism through which we are able to investigate the model's ability to deal with varying positional scales as we do in section A.5 of the Appendix. Through this we also present an additional reason why architectures that utilize RoPE do not usually generalize beyond the sequence lengths they were trained on, a result that has implications beyond just RoMAE. We will add additional discussion on this matter to the manuscript.
> > > >
> > > > Regarding your sentiment toward the versatility of RoMAE, we would like to provide a possible middle ground summarised by the following sentence that we will add to the manuscript: “RoMAE is to irregular time series what ViT was to images: a minimal Transformer recipe anyone can copy-paste for interpolation/classification tasks”. The versatility of RoMAE lies in its ease of acces as a go-to model for these kinds of tasks.

---

### Official Review · Reviewer_wtxL · 2025-07-03

**Clarity:** 3
**Significance:** 2
**Originality:** 3
**Rating:** 5
**Confidence:** 3

**Summary:**

The Rotary Masked Autoencoder (RoMAE) adds a number of positional encoding improvements into Masked Autoencoders, studying in detail the variations of behavior that the resulting positional encodings allow. In a broad series of experiments across regular and irregular time-series, images, and audio, RoMAE is compared to existing baselines, quantified empirically, and studied technically to show the benefits of adding positional information into the MAE framework. This results in a simplification and unification in terms of architecture, compared to baselines in the respective domains studied.

**Questions:**

Related to weaknesses above, can you clarify a bit more the exact application of the continuous axial RoPE to multivariate, irregular timeseries? This seems intricate but critical for a number of areas, and a potential strength of RoMAE.

On infilling tasks, do you notice any particular bias in the model toward low or high frequency components? Most of the examples shown seem to have low frequency patterns, but showing that the model is also good at dealing with high-frequency, repeating ones would be useful for many applications.

In the limitations section, it is briefly mentioned that "RoPE in RoMAE has some additional computational overhead if the positions are
336 different with each forward pass, e.g., with any continuous irregular time-series". This seems like a very significant drawback to me - as the speed of RoPE application can sometimes drives technical modeling decisions. Are there any empirical studies around this speed difference, compared to for example a model with no RoPE at all? Or one that receives the continuous position value directly as an additional input?

There are numerous studies around the interpolation of RoPE parameters for context expansion (YaRN for example). The interpolation section didn't mention much, if any, about these methods but they seem directly applicable for things like training on lower resolution images, and applying to higher resolution ones, or downsampled timeseries training, etc. Have the authors looked into this, or have some clarification about potential applications here?

**Ethical Concerns:**

["NO or VERY MINOR ethics concerns only"]

**Final Justification:**

The authors addressed a number of my questions, and the promised revisions should strengthen the paper. Given the updates, I think the paper is acceptable for the conference, but without application to categories of broader interest might be somewhat niche in its appeal - particularly the extrapolation aspect discussed by the other reviewers, as well as hinted at in my own review, is very important and is key within time-series applications. That said, the interpolation ability and simplicity of this approach make it appealing for a large number of time-series tasks as well. I do think it is a reasonably strong paper for the niche, once the focus on clarifying details has been added. Particularly, the author replies to reviewer questions has been thorough and answered some important questions around the method.

**Limitations:**

Yes

**Paper Formatting Concerns:**

No major concerns.

**Quality:**

3

**Strengths And Weaknesses:**

Strengths:
This paper features extensive empirical findings, experimental results, and extensions into a broad appendix with many interesting findings. The discussions about the subtlety of including the [CLS] token are practically relevant, and Figure 1 is a great introduction to the method, and the overall early description and background are clear. The results make sense and are compelling for the domains, many sections help the reader understand the work, and the RoMAE architecture seems useful and potentially simple to reproduce for new applications. The work seems broadly applicable to many areas given the generality of the MAE architecture, and the authors do a good job of conveying the various configurations of RoMAE that may be applicable to a given application area.

Weaknesses:
One drawback from the broad swath of experimental and technical findings here, is that the paper flow feels a bit convoluted. I would strongly encourage the authors to re-visit the narrative structure of the paper, and re-organize. Delegating a number of core results tables to the appendix, while also having a half-page dedicated to irregular time-series, and numerous tables surrounded by empty white space, shows that a deep-clean in terms of paper format can both increase the information density of the paper, and also improve the flow for interested readers. Details about dataset construction are more appropriate for appendix material (beyond brief discussion in the main paper), rather than core findings.

The description of how to apply continuous axial RoPE to irregular time-series is a bit unclear to me, and would be greatly boosted by a figure showing the irregular time-series processing. Section 4.3 seems to take up a lot of space in the paper to describe a simple (but useful) finding, and might do with some changes to compress the information.

Given the core of MAE is fundamentally about representation learning, some qualitative exploration of learned representations could be a useful bridge between the detailed discussion of CLS token and absolute position, and dataset specific benchmarks in the later sections. Similar experiments were done in both VQ-VAE and MAE using linear probing, and may be useful to guide ideas in this direction.

A second drawback from the breadth of experiments, is that the comparisons (generally) lack depth. It may be out of scope for this revision round, but consider a more direct application of this approach to "standard" benchmarks, adding direct classification scoring to the Tiny ImageNet results and adding in numerous relevant benchmarks, adding these representations as grafted representations to a large language model for multimodal tasks (similar to approaches like miniGPT, LLaVa, Freeze-Omni, and so on), or other benchmarks (direct comparisons to ViT, even if a small one, on some standard tasks). As it stands, the existing benchmarks seem time-series focused, but time-series model benchmarking usually covers an even larger set of datasets, so the paper currently sits in the middle of two camps, and stronger multimodal benchmarking would help me improve the score if that is the preferred direction for the authors.

I do not find a key "hero result" to latch onto here, in a similar spirit to the (well-done) "hero" Figure 1. Generally, a lot of time is devoted to the details of absolute position prediction, which is useful but some portion of that discussion may be better suited to the appendix.

---

> ### Author Rebuttal · Authors · 2025-07-31
>
> We thank the reviewer for their time and their useful feedback. We have taken on board several points highlighted in the review that we believe will strengthen our paper.
> Let us first address the questions:
>
> ## Question 1:
> When dealing with multivariate irregular time series, we distinguish 2 cases depending on the type of irregularity of the data:
>
> a) All the features present the same irregularity pattern (in time), meaning that all variates (features) are observed at each available time-step,
>
> b) Several axes are irregular (e.g. we observe one variate at a time step, then another at a different time-step),
>
> In case a),  we utilize patchification to create one embedding combining all variates for each available time-step, embedding only time through the application of 1D p-RoPE.
>
> When dealing with case b), in practice, we flatten the irregular dimensions into a 1D sequence, saving their original “dimensional index” in their positional embedding. Any features present at *all* time-steps can either be encoded in additional positional dimensions, e.g., the time attribute could be encoded in its own dimension as every point has a time value, *or* through patchification. In the case of ELAsTiCC, we use two Continuous Axial RoPE dimensions; one storing the time of each token, and the other storing the index of the variate, with the value of each point being inserted through patchification.
>
> We also point out that in case b), our novel theoretical analysis of absolute position reconstruction was required  to show that the model is able to deduce the “dimensional index” of each point. We are working on an additional Figure to make the irregular multi-positional encoding process clearer in the paper, along with improving the description in the text.
>
> ## Question 2:
> This is a very interesting question. Given that our model is trained using MSE, we hypothesize that it might suffer from the “spectral bias” towards prioritizing low frequencies (Rahaman et al, 2019 “On the Spectral Bias of Neural Networks”). However, and as highlighted in Rahaman et al., the picture might not be that straightforward depending on the data shape and the problem (see Sec 5 in their paper). Therefore, from a purely theoretical point of view, it is not trivial to give a clear-cut answer to this question.
>
> To investigate more empirically if RoMAE is also good at handling high-frequency patterns, we are currently running experiments on a controlled toy dataset mixing sine-waves at different frequencies to investigate if higher frequencies are well retained during interpolation; this will be integrated in the revised manuscript.
> It is also relevant to note that the ELAsTiCC dataset used in our Section 5.3 experiment contains a good diversity of frequency patterns; It is a dataset where each class corresponds to a type of astrophysical object with specific variability patterns and frequencies (some recurring, some not). RoMAE’s good performance on this dataset would thus tend to indicate that it is robust for various frequencies.
>
> ## Question 3
> We agree that evaluating the additional overhead incurred by continuous RoPE for irregular sequences is important. We would like to firstly point out that “Rotary Position Embedding for Vision Transformer” (Heo et al. 2403.13298)​​ showed that regular RoPE adds negligible additional cost to the overall computational cost. In particular, they showed that during inference, injecting 2-D RoPE into ViT-B adds 1.8 M FLOPs, just 0.01 % of the model’s total GFLOPs ( 17.6 G FLOPs).
> We note that in general, positional embeddings for irregular time-series cannot be cached, as they change with every forward pass, which makes RoPE on irregular time-series slower than regular RoPE.
> We have performed a quick test demonstrating relative speed up for a typical image (224x224) workload. The results are as following:
> | Model | Relative speed (Baseline = MAE)|
> |-------------|-------------:|
> | MAE  | 1 |
> | RoMAE (regular positions)  | 0.98  |
> | RoMAE (irregular positions)  | 0.87  |
>
> ## Question 4:
> Yes, using context expansion (for example as with YaRN) would absolutely be possible here, e.g. for images (as mentioned by the reviewer, training on lower-resolution and evaluating on higher resolution). We will add a sentence on that regard in the manuscript.
>
> ## General comments
> **Hero result**: We would like to point out that our results on the ELAsTiCC dataset would constitute a remarkable result, given that RoMAE is the best performing model on this dataset (to our knowledge) which is of great interest within the astrophysics community. We therefore envisage follow-up studies, using models like RoMAE on real data, will be possible when real data from the LSST survey is released. The fact that analysis on such a dataset is even tractable, let alone feasible, should motivate readers to consider this result our ‘hero’ result. We did not emphasise this in the manuscript due to the scope of the conference, but , given your comment, we understood that a little more context is maybe needed for the camera ready version.
> We also want to acknowledge the recommendations on the manuscript’s organization to improve its clarity and structure; we will restructure by moving the patchification subsection to the Appendix, collating Tables of results and moving the audio result in the main part, and will add an illustration for the positional embeddings in the multi-variate irregularly sampled case.
>
> We also appreciate the comment regarding a qualitative analysis of the learned representations, using e.g. the linear probing. We agree that it is relevant for a variety of applications. We are currently running linear probing experiments on our TinyImageNet embeddings to compare the effect of CLS and absolute-position encoding, which will be included in the revised version.
>
> We also agree that investigating the integration of RoMAE in multi-modal frameworks would be interesting and particularly relevant for e.g. applications in astronomy (related to the ELAsTiCC dataset), among others. However, we feel such an analysis would be out of the scope of the current manuscript (and realistically not possible for us to conduct properly within this round of rebuttal as acknowledged by the reviewer); we hope nonetheless that the breadth of experiments conducted (various settings, nature of the data and problems) are sufficient to showcase the advantages (and limitations) of RoMAE and motivate future works investigating multi-modal settings.

---

> > ### Comment · Reviewer_wtxL · 2025-08-05
> > **Reply to Authors**
> >
> > Thank you to the authors for the detailed response.
> >
> > These answered all of my core questions, and with the additional promised changes it should help conference readers at large understand the approach as well as the impact of this work.
> >
> > More discussion of the impact of the ELAsTiCC results in the paper text itself will definitely help readers (like me) who are not familiar with the task, to better understand the impact of performance here. After the discussion, I see how it is a standout result and demonstrates the flexibility of the method.
> >
> > While strictly optional, I hope the authors will also consider open sourcing code (or pseudo-code for the core approach in the appendix), as I think it could have a large impact and downstream use - due to the simplicity and effectiveness of the approach.

---

> > > ### Author Response · Authors · 2025-08-07
> > >
> > > We are glad we could successfully address your questions.
> > > We will definitely make the code open source.

---

### Public Comment · ~Yiming_Ma3 · 2026-03-22
**Overclaim on Vision Results Without a True MAE Baseline**

The paper overclaims on its vision results. It asserts that RoMAE preserves MAE-level performance on images, but never actually reports a true MAE baseline. Replacing MAE with a vaguely “MAE-like” absolute-position variant and then treating the result as evidence against MAE is not a valid empirical comparison. This is not a minor presentation issue; it directly affects one of the paper’s headline claims. As written, the image-domain conclusion is unsupported and should be considered unsubstantiated until the authors provide a proper controlled comparison against standard MAE. The fact that reviewers did not explicitly call this out suggests that this part of the paper was not rigorously stress-tested during peer review.

---

### Note · Authors · 2025-08-12

We would like to first thank all reviewers for their useful comments, suggestions, and questions. We are encouraged by the generally positive responses to our rebuttals. Below, we summarize the changes and additions that will be made to the final version of the manuscript:

- A new figure illustrating the process of applying RoPE for multi-dimensional continuous position, as well as formatting improvements (Reviewer wtxL)
- A controlled experiment with a toy dataset mixing sine-waves at different frequencies to empirically investigate if higher frequencies are well retained during interpolation. (Reviewer wtxL)
- Direct estimation of the model computational cost (in different settings) and comparisons with Latent ODE and HeTVAE. (Reviewers wtxL, joLj, Wjq7)
- The addition of a clarification regarding RoMAE’s importance for irregular time-series studies, along with a qualitative remark regarding the parallels with ViT and its usage with images. (Reviewer mZNG)

We believe these additions will further strengthen the clarity, completeness, and impact of the work.

---

### Decision · Program_Chairs · 2025-09-17

**Decision:**

Accept (poster)

**Comment:**

This paper improves the positional encoding of Masked Autoencoders to handle time-series data and evaluates them in a range of domains and against existing baselines. The feedback for this paper had been mixed, with reviewers arguing that the scope of the problem and the novelty are moderate, as well as with questions around handling extrapolation. During the discussion period, the authors addressed some of these questions and provided a thorough rebuttal to some of these questions. Overall, I recommend acceptance of this paper.